



# Dithiothreitol Activity by Particulate Oxidizers of SOA Produced from Photooxidation of Hydrocarbons under Varied NO$_x$ Levels

Huanhuan Jiang,[1] Myoseon Jang,[1] Zechen Yu,[1]

[1]Department of Environmental Engineering Sciences, Engineering School of Sustainable Infrastructure and
Environment, University of Florida, Gainesville, FL 32608, USA

*Correspondence to*: Myoseon Jang (mjang@ufl.edu)

**Abstract.** When hydrocarbons are atmospherically oxidized, they form particulate oxidizers, including quinones, organic hydroperoxides, and peroxyacyl nitrates (PANs). These particulate oxidizers can modify cellular materials (e.g., proteins and enzymes), and adversely modulate cell

functions. In this study, the contribution of particulate oxidizers in secondary organic aerosols (SOA) to the oxidative potential was investigated. SOA were generated from the photooxidation of toluene, 1,3,5-trimethylbenzene, isoprene, and α-pinene under varied NO$_x$ levels. Oxidative potential was determined from the typical mass-normalized consumption rate (reaction time $t = 30$ min) of dithiothreitol (DTT$_t$), a surrogate for biological reducing agents. At high NO$_x$ conditions,

the DTT$_t$ of toluene SOA was 2–5 times higher than that of other types of SOA.  Isoprene DTT$_t$ significantly decreased with increasing NO$_x$ (up to 69 % reduction by changing the hydrocarbon/NO$_x$ ratio from 30 to 5). The DTT$_t$ of 1,3,5-trimethylbenzene and α-pinene SOA was insensitive to NO$_x$ under the experimental conditions of this study. The significance of quinones to the oxidative potential of SOA was tested through the enhancement of DTT consumption in the

presence of 2,4-dimethylimidazole, a co-catalyst for the redox cycling of quinones; however, no significant effect of 2,4-dimethylimidazole on modulation of DTT consumption was observed for all SOA, suggesting that a negligible amount of quinones was present in SOA of this study. For toluene and isoprene, mass-normalized DTT consumption (DTT$_m$) was determined over an extended period of reaction time ($t = 2$ h) to quantify their maximum capacity to consume DTT.

The total quantities of PANs and organic hydroperoxides in toluene SOA and isoprene SOA were also measured using the Griess assay and the 4-nitrophenylboronic acid assay, respectively. The amount of organic hydroperoxides was substantial, while PANs were found to be insignificant for both SOA.  Isoprene DTT$_m$ was almost exclusively attributable to organic hydroperoxides, while toluene DTT$_m$ was partially attributable to organic hydroperoxides.  The results of the model

compound study suggest that electron-deficient alkenes, which are abundant in toluene SOA, could also modulate DTT$_m$.



# 1 Introduction

Epidemiological studies have linked human exposure to fine particulate matter (PM$_{2.5}$, aerodynamic diameter < 2.5 μm) to increased morbidity and mortality from respiratory and cardiovascular diseases (e.g., asthma, myocardial infarction, stroke) (Brook et al., 2010; Chen et al., 2013; Davidson et al., 2005; Jansen et al., 2005; Katsouyanni et al., 1997; van Eeden et al., 2005). Primary combustion particulates (e.g., wood smoke particles, vehicle emissions) are known to be causative agents of these diseases (Danielsen et al., 2011; Nel, 2005); however, increasing attention is also being paid to secondary organic aerosols (SOA) (Jang et al., 2006; Lin et al., 2016; McDonald et al., 2010), which are produced from the atmospheric transformation of hydrocarbons (HCs) in the presence of atmospheric oxidants (e.g., NO$_x$, OH radicals, O$_3$) (Hallquist et al., 2009). Although SOA comprises a large fraction of PM$_{2.5}$ (20–90 %) (Gelencsér et al., 2007; Kanakidou et al., 2005), its mechanistic role in causing adverse health effects remains unclear.

The toxicity of organic aerosols has been ascribed to the generation of reactive oxygen species (ROS) and the modification of biomolecules (e.g., DNA and cellular enzymes) (Danielsen et al., 2011; Nel, 2005). ROS can induce oxidative stress in pulmonary systems, followed by a cascade of inflammation responses, and ultimately the apoptosis of lung cells (Danielsen et al., 2011; Li et al., 2003; Li et al., 2008). Particulate organic compounds such as quinones and polyaromatic hydrocarbons can react with cellular reducing agents (e.g., NADPH) and form ROS (i.e., H$_2$O$_2$ and O$_2^{\cdot -}$) (Kumagai et al., 2012). To efficiently determine the oxidative potential (the ability to generate ROS) of different types of particulate matter at a laboratory benchtop scale, a low-cost acellular technique, dithiothreitol (DTT) assay, has been widely used (Antiñolo et al., 2015; Cho et al., 2005; Hedayat et al., 2014; Janssen et al., 2014; Kramer et al., 2016; Verma et al., 2015). DTT acts as a surrogate for biological reducing agents owing to its two sulfhydryl groups. Some quinones (e.g., 1,4-naphthoquinone (NQN) and 9,10-phenathraquinone (PQN)) can efficiently consume DTT via a catalytic redox cycle, during which quinones are reduced to semiquinones or hydroquinones (Chung et al., 2006; Li et al., 2003). Hence, quinone compounds, commonly found in primary combustion particulates (Danielsen et al., 2011; Jakober et al., 2007), are known to be important contributors to DTT response of combustion particles.

Unlike combustion PM, biogenic SOA and most aromatic SOA (except naphthalene SOA) contain little or no quinones (Forstner et al., 1997; Hamilton et al., 2005; McWhinney et al., 2013; Pindado Jiménez et al., 2013); however, recent work has shown that the DTT activity of SOA (toluene,





1,3,5-trimethylbenzene (TMB), isoprene, and α-pinene) was high and even comparable to that originating from combustion particulates (e.g., wood smoke particles) (Jiang et al., 2016), suggesting that there must be unidentified mechanisms underlying DTT consumption other than the catalytic act of quinones.

In this study, three groups of SOA products were introduced to explain the mechanistic role of SOA products in DTT consumption (Fig. 1a and b). First, non-catalytic particulate oxidizers in SOA, such as organic hydroperoxides (alkyl hydroperoxides and acyl hydroperoxides) and peroxy acyl nitrates ($RC(O)OONO_2$; PANs), can oxidize sulfhydryl groups in DTT to form disulfides, sulfenic acids ($RSOH$), sulfinic acids ($RSO_2H$), or sulfonic acids ($RSO_3H$) (Grek et al., 2013;

Mudd, 1966). These non-catalytic particulate oxidizers are abundant in SOA sourced from various hydrocarbons (Docherty et al., 2005; Sato et al., 2012). Second, catalytic particulate oxidizers, such as quinoid substances, can oxidize sulfhydryl groups through a redox cycle (Cho et al., 2005; Kumagai et al., 2002). A trace amount of quinones can be found in aromatic SOA products (Forstner et al., 1997). Third, electron-deficient alkenes in SOA can react with the sulfhydryl

groups of DTT via a Michael addition (Nair et al., 2014). Alkenes substituted with an electron withdrawing group (e.g., conjugated carbonyls) are commonly found in ring-opening products from the photooxidation of aromatic HCs (e.g., toluene) (Jang and Kamens, 2001; Saunders et al., 2003, 1997; Wyche et al., 2009). The contributions of all three groups of SOA products to DTT activity can be influenced by the type of precursor HC (aromatics vs. biogenics) and by $NO_x$

($NO+NO_2$) levels (HC/$NO_x$ ratios) (Eddingsaas et al., 2012b; Jang and Kamens, 2001; Wyche et al., 2009; Xu et al., 2014).

  Advanced analytical instruments (e.g., aerosol mass spectrometers and liquid chromatograph mass spectrometers integrated with soft ionization) have innovated the characterization of SOA compositions; however, their data are limited to elemental analysis (Xu et al., 2014) or the

identification of some chemical species (e.g., carboxylic acids and carbonyls) by a unique fragmentation (Sato et al., 2012; Shiraiwa et al., 2013). Particulate oxidizers (e.g., PANs and organic hydroperoxides) are thermally unstable and can decompose during chemical injection at high temperature, making it difficult to characterize SOA compositions using mass spectrometers (Zheng et al., 2011). This difficulty is also compounded by a lack of authentic standards suitable

for the analysis of diverse and complex particulate oxidizers.





The purpose of this study is to characterize the effect of SOA products on DTT consumption. SOA were generated from the photooxidation of different HCs under varied environmental conditions ($NO_x$ levels) using a large outdoor photochemical smog chamber. The two most abundant anthropogenic HCs (i.e., toluene and TMB) in the ambient atmosphere and the two ubiquitous biogenic HCs (i.e., isoprene and α-pinene) were chosen as SOA precursors. Aerosols were collected using an online technique with a particle-into-liquid sampler (PILS). SOA samples were immediately applied to the DTT assay and the quantification of particulate oxidizers. The amount of PAN was measured using the Griess assay and that of organic hydroperoxides was measured using the 4-nitrophenylboronic acid (NPBA) assay. The contribution of quinones to the oxidative potential of SOA was estimated by the enhancement of DTT consumption in the presence of 2,4-dimethylimidazole, a co-catalyst for the redox cycling of quinones (Dou et al., 2015). In addition to particulate oxidizers, the contribution of electron-deficient alkenes to DTT activity was investigated for aromatic SOA (toluene SOA). Although the chemical assays (e.g., NPBA assay and Griess assay) used in this study have limitations (e.g., providing structural details of organic compounds), they are user-friendly and can accurately quantify the total amount of organic hydroperoxides and PANs, both of which are important for understanding the role of SOA in cellular oxidative stress at the molecular level. The quality control (QC) of the chemical assays used in this study will be discussed.

## 2 Materials and methods

### 2.1 Outdoor chamber experiments

SOA were generated under natural conditions (ambient sunlight, temperature, and relative humidity) using the University of Florida Atmospheric PHotochemical Outdoor Reactor (UF-APHOR) dual chambers (52 $m^3$ each). Before each experiment, the chambers were flushed with the clean air for 2 days using an air purifier system (GC Series, IQAir Inc.) until the background particle mass concentration was below 1 μg $m^{-3}$. HC and $NO_x$ were injected to the chamber before sunrise. For photooxidation experiments of toluene, HONO generated from the reaction of 0.1 M $NaNO_2$ solution and 10 % w/w $H_2SO_4$ solution was injected into the chamber as a source of OH radicals. HONO produced OH radicals and NO via photolysis. The particle size distribution of chamber SOA was monitored using a scanning mobility particle sizer (SMPS), and was converted





to the mass concentration using the SOA density (1.3 g cm$^{-3}$ for α-pinene SOA and 1.4 g cm$^{-3}$ for the other types of SOA) (Ng et al., 2007a; Ng et al., 2007b; Wyche et al., 2009; Xu et al., 2014). SOA were generated under varied NO$_x$ conditions (HNOX: high NO$_x$, LNOX: low NO$_x$; Table 1). Other details about chamber experiments can be found in Sect. S1 of the Supporting Information.

## 2.2 Sampling method

SOA and background (before chemical injection) samples were collected within a small amount of deionized (DI) water using a PILS technique equipped with an upstream parallel carbon filter denuder (Sunset Laboratory Inc.) to remove gaseous compounds. The efficiency of the carbon denuder was measured by comparing the concentrations of toluene and CCl$_4$ with the carbon denuder to those without the denuder and was found to be almost 100 %. The sampling efficiency of PILS is higher than 85 % for particles larger than 0.1 μm (Orsini et al., 2003). The mass concentration of SOA in the PILS sample was estimated using the chamber SOA mass concentration, the air flow rate of PILS, the total liquid volume collected by PILS, and the collection efficiency of PILS. SOA samples collected by PILS were applied to the chemicals assays described in Sect. 2.3.

To measure the concentration of PANs in the gas phase, gaseous photooxidation products (23 May 2016) were collected using an impinger (filled with 5 mL DI water) at a flow rate of 0.8 L min$^{-1}$. A filter (13 mm diameter, Pall Life Scientific Pallflex, TX40HI20-WW) was applied upstream of the impinger to remove particles. The impinger samples were then applied to PAN analysis.

## 2.3 Chemical assays

Detailed information about chemicals and solution preparation can be found in Sect. S2. To avoid the decay of some unstable SOA products in the aqueous solution, the analytical procedures of DTT, PAN, and organic hydroperoxides assays were completed within 24 h after sampling. Before chemical analysis, all SOA samples were stored in a refrigerator at −4 °C. Background chamber air samples and blank controls (DI water) were applied to all chemical assays for each set of measurements. The limit of detection of each chemical assay is shown in Table S1.

### 2.3.1 DTT assay

DTT assay (two steps) was employed to quantify the oxidative potential of SOA (Cho et al., 2005; Jiang et al., 2016). In the first step, a mixture of 700 μL SOA sample, 200 μL potassium phosphate





buffer (2 mM) and 100 µL DTT (1 mM) was incubated at 37 °C, and shaken in a sonicator (FS30H Ultrasonic Cleaner, Fisher Scientific). For the second step, the reaction between DTT and SOA was quenched by adding 1 mL trichloroacetic acid (1 % w/v). Then, 0.5 mL 5,5'-dithiobis-(2-nitrobenzoic acid) solution (1 mM in methanol) was added to react with the remaining DTT

forming 2-nitro-5-thiobenzoic acid, which produced a yellow color after the addition of 1 mL Tris base buffer (pH = 8.9, 0.4 M). The absorbance of 2-nitro-5-thiobenzoic acid at 412 nm was measured using a UV/VIS Spectrometer (Lambda 35, PerkinElmer). Positive controls (0.1 µM PQN) were run in duplicates for each set of measurements. The blank-corrected DTT consumption ($\Delta$DTT, nmol) was estimated by Eq. (1):

$$\Delta\text{DTT} = \frac{A_{blk} - A_{SOA}}{A_0} \text{DTT}_0 \tag{1}$$

where $A_{blk}$ is the absorbance of the blank control after incubation, $A_{SOA}$ is the absorbance of the SOA sample after incubation, $A_0$ is the absorbance of the blank control without incubation, and $\text{DTT}_0$ (100 nmol) is the initial moles of DTT. As shown in Fig. S1, the DTT loss rate in blank control during sonication (0.162 nmol min$^{-1}$) was close to that during shaking (0.153 nmol min$^{-1}$,

shaken by an Edison Environmental Incubator Shaker G24, low speed, 37 °C); therefore, in this study, the influence of free radicals generated by sonication on DTT measurement was insignificant. The SOA mass applied to DTT assay was constrained to ensure that the DTT consumption remained less than 50 % of $\text{DTT}_0$.

### 2.3.2 Organic hydroperoxides analysis

The NPBA method, which had been used by Su et al. (2011) for the determination of $H_2O_2$, was extended for the quantification of alkyl and acyl hydroperoxides. NPBA reacts with organic hydroperoxides to form a 4-nitrophenol (Scheme S1), which has a large absorption coefficient at 406 nm (Kuivila, 1954; Kuivila and Armour, 1957; Su et al., 2011). A mixture of 1 mL SOA sample, 100 µL NPBA solution (10 mM in methanol), and 900 µL KOH solution (50 mM) was

incubated at 85 °C. The absorbance at 406 nm of SOA sample was measured before reaction with NPBA and was found to be negligible. Positive controls (10 µM $H_2O_2$) were run in duplicate for each set of measurements. It has been reported that boronic acid can react with multi-alcohols to form colorful products (Kim et al., 2007). To examine the possible interference by NPBA-alcohol adducts, the glycerol aqueous solution was tested using the NPBA method, but no measurable

absorption appeared in UV spectrum. Therefore, the UV absorption spectrum of this study



originated from 4-nitrophenol, products from the reaction of NPBA with organic hydroperoxides in SOA. No more than 10 µg SOA was applied to the NPBA assay. For toluene SOA, the reaction of organic hydroperoxides with NPBA completed within 6 h, and for isoprene SOA within 2 h (Fig. S2). The NPBA method was calibrated using aqueous 4-nitrophenol solutions ranging from

1–40 µM (Fig. 2a). The feasibility of the NPBA assay was tested for peracetic acid ($CH_3C(O)OOH$), tert-butyl peroxide ($(CH_3)_3COOC(CH_3)_3$), tert-butyl hydroperoxide ($(CH_3)_3COOH$), and hydrogen peroxide ($H_2O_2$). Within a 90 % confidence level, the absorbance sourced from the reaction of NPBA with the known amount of organic hydroperoxides or $H_2O_2$ was covered by the calibration curve (Fig. 2a). However, no absorbance at 406 nm appeared in the

NPBA+tert-butyl peroxide mixture (data not shown here). Organic hydroperoxides in aqueous solution are unstable. For example, after one week in a refrigerator at 4 °C, $(CH_3)_3COOH$ was slightly degraded; therefore, we ensured that SOA samples were applied to chemical assays soon after collection (within 24 h).

### 2.3.3 PAN analysis

The concentration of PANs was quantified by Griess assay. The Griess reagent, a mixture of sulfanilic acid and n-(1-naphthyl) ethylenediamine dihydrocloride (NEDD), has been widely applied to quantify the concentration of nitrogen oxides in environmental, industrial, and biological systems (Giustarini et al., 2008; Ridnour et al., 2000; Saltzman, 1954). Nitrogen oxides were transformed to nitrites that can form azo dyes when mixed with Griess reagent (Scheme S2)

(Giustarini et al., 2008). For PAN analysis, a 300 µL SOA (collected by PILS) or gas sample (collected by an impinger) was mixed with 300 µL KOH aqueous solution (50 mM) for 15 min to hydrolyze PANs completely and form nitrites. Then, 1 mL Griess reagent (20 mM sulfanilic acid and 5 mM NEDD aqueous solution) was added to the mixture, and allowed to react with nitrites for 15 min. A purple-magenta color formed immediately. The concentration of PANs was

estimated from the absorbance at 541 nm (Ridnour et al., 2000). No difference in the absorbance was found between a 15min reaction and a 30min reaction. Positive controls (10 µM $NaNO_2$) were run in duplicate for each set of measurements. Griess assay was calibrated using $NaNO_2$ aqueous solutions with varied concentrations (0.4–50 µM; Fig. 2b).





### 2.3.4 DTT activity enhancement

Dou et al. (2015) showed that by forming H-bonds with hydroquinones, imidazole derivatives are capable of facilitating electrons transfer from hydroquinones to molecular oxygen, accelerating the redox cycling of quinones and enhancing the oxidation of DTT. In the DTT enhancement test,

a 250 μL 2,4-dimethylimidazole aqueous solution (5 mM) was mixed with a 450 μL SOA-PILS sample to get a 700 μL mixture. Then, 100 μL DTT solution (1 mM) and 200 μL potassium phosphate buffer (2 mM) were added to the mixture. The subsequent steps were the same as those used for the DTT assay. The enhanced DTT consumption rate ($t = 30$ min) in the presence of 2,4-dimethylimidazole was measured. Positive controls (2 μM NQN) were run in duplicates for each

set of measurements.

### 3 Results and discussion

### 3.1 DTT activity of SOA.

The SOA yield ($Y$) represents a ratio of organic mass formed to HC consumed (Odum et al., 1996). As shown in Table 1, the $Y$ values of toluene, TMB, isoprene, and α-pinene SOA ranged from 5

% to 25 %, 6 % to 8 %, 1 % to 5 %, and 14 % to 36 %, respectively. The trends in SOA yields, regarding to $NO_x$ levels and the HC type, were consistent with those reported in previous chamber studies (Eddingsaas et al., 2012a; Im et al., 2014; Jiang et al., 2016; Kroll et al., 2006; Xu et al., 2014).

The DTT consumption rate, $DTT_t$ (pmol min$^{-1}$ μg$^{-1}$), was defined as DTT consumption (ΔDTT,

pmol) per minute of reaction time ($t$, min) per microgram of SOA mass ($m_{SOA}$, μg):

$$DTT_t = \frac{\Delta DTT}{m_{SOA}t} \tag{2}$$

Figure 3 illustrates the $DTT_t$ ($t = 30$ min) of SOA produced from four different HCs under varied $NO_x$ conditions. Overall, the influence of $NO_x$ on $DTT_t$ varied, depending on the type of HC. The $DTT_t$ of toluene SOA was insensitive to $NO_x$ for samples collected within a similar sampling

period, but it decreased with increasing aging time. The $DTT_t$ of toluene SOA reached approximately 70 pmol min$^{-1}$ μg$^{-1}$ by 13:00 under both high $NO_x$ and low $NO_x$ conditions, but decreased by about 40 % in the late afternoon. For aged toluene SOA, the decline of $DTT_t$ might reflect the decay of photooxidation products that could potentially react with DTT (e.g., electron-deficient alkenes can react with ozone and OH radicals) (Finlayson-Pitts and Pitts Jr, 2000). The





lifetime of acrolein (an electron-deficient alkene), is 3 h in the presence of OH radicals ($2.0 \times 10^{-4}$ ppb) and 208 h in the presence of ozone (200 ppb) (Sect. S7) (Finlayson-Pitts and Pitts Jr, 2000; Saunders et al., 2003, 1997). Furthermore, some particulate oxidizers might also photochemically decompose with increasing oxidation time (see Sect. 3.3). For isoprene SOA, $DTT_t$ was

significantly affected by $NO_x$. There was a 38 % to 69 % decrease in isoprene $DTT_t$ when the $HC/NO_x$ ratio was reduced from 30 to 5. Under high $NO_x$ conditions, the $DTT_t$ of less-aged isoprene SOA was about 50 % lower than that of less-aged toluene SOA. However, under low $NO_x$ conditions, the $DTT_t$ of isoprene SOA was comparable to that of toluene SOA. The $DTT_t$ of TMB and α-pinene SOA were much lower than those of toluene and isoprene SOA, and they were not

affected significantly by $NO_x$ conditions. Fujitani et al. (2012) measured the expression of heme oxygenase 1 (HO-1) in *in vitro* mouse airway epithelial cells exposed to three different SOA sourced from the ozonolysis of α-pinene and the photooxidation of TMB or m-xylene in the presence of $NO_x$ (low $NO_x$ levels). In their study, the HO-1 expression in cells exposed to aromatic SOA was significantly higher than that in cells exposed to α-pinene SOA, which is consistent with

the trend in $DTT_t$ found in this study.

Traditional $DTT_t$ has been used to measure the oxidative potential originating from the catalytic redox cycling of particulate constituents (e.g., quinones and metals) (Charrier and Anastasio, 2012; Cho et al., 2005; Kumagai et al., 2002). When governed by such catalytic reactions, DTT consumption increased linearly with reaction time (Fig. S3). To demonstrate the time-dependency

of DTT consumption, the reaction time of DTT assay was extended to 2 h for isoprene SOA and toluene SOA, which both had high $DTT_t$. The mass-normalized DTT consumption ($DTT_m$, nmol $\mu g^{-1}$) was defined as the ratio of $\Delta DTT$ (nmol) to $m_{SOA}$ (μg):

$$DTT_m = \frac{\Delta DTT}{m_{SOA}} \qquad (3)$$

In Fig. 4, the $NO_x$ effect on $DTT_m$ was consistent with that on $DTT_t$: no $NO_x$ effect was observed

on the $DTT_m$ of toluene SOA, and the $DTT_m$ of low-$NO_x$ isoprene SOA was much higher than that of high-$NO_x$ isoprene SOA.

Figure 4 shows that the increase in $DTT_m$ with time for both isoprene and toluene SOA was non-linear, suggesting that DTT consumption by SOA products was governed by non-catalytic processes. For example, DTT consumption by isoprene SOA was nearly completed within 2 h.

For toluene SOA (Toluene HNOX and Toluene LNOX), the increase of $DTT_m$ also appeared to slow down over a 2h reaction, although the $DTT_m$ did not reached a plateau under the same DTT





assay conditions (i.e., the same buffer concentration). Medina-Ramos et al. (2013) reported that the electron transfer rate between glutathione (GSH) and an electrogenerated mediator ($[IrCl_6]^{2-}$) exhibited a slight acceleration when the phosphate buffer concentration was increased from 0 to 50 mM at pH = 7.0. To achieve the completion of the reaction between particle oxidizers in SOA

and DTT, the $DTT_m$ of toluene SOA (Toluene LNOX-17 Nov 2016) was measured with a 0.8 mM potassium phosphate buffer in the first step of DTT assay (2 times higher than the typical buffer concentration (0.4 mM)). As shown in Fig. 4, the $DTT_m$ of the toluene SOA (LNOX-17 Nov 2016) reached a plateau within 2 h, proving that DTT consumption by toluene SOA was controlled by non-catalytic mechanisms. At high $NO_x$ conditions, the $DTT_m$ ($t$ = 2 h) of toluene SOA was 4–5

times higher than that of isoprene SOA. This difference was about 2 times greater than that for $DTT_t$ (Fig. 3); therefore, we concluded that $DTT_m$ is more suitable than $DTT_t$ for estimating the oxidative potential of SOA, given that $DTT_m$ can determine the maximum capacity of non-catalytic modulators in SOA to consume DTT.

### 3.2 DTT modulator: quinones

To illustrate the role of quinones in modulating the DTT responses of SOA, the enhanced DTT consumption rate ($t$ = 30 min) in the presence of 2,4-dimethylimidazole was measured. The enhancement factor (pmol $min^{-1}$ μg-$SOA^{-1}$ μmol-$imidazole^{-1}$) was estimated by Eq. (4):

$$\text{Enhancement Factor} = \frac{\Delta DTT_{mix} - \Delta DTT_{SOA} - \Delta DTT_{imidazole}}{m_{SOA}\, n_{imidazole}\, t} \qquad (4)$$

where $n_{imidazole}$ (μmol) is the moles of 2,4-dimethylimidazole added to the DTT reaction mixture,

$\Delta DTT_{mix}$ (pmol) is the DTT consumption by the mixture of SOA and 2,4-dimethylimidazole, $\Delta DTT_{SOA}$ (pmol) is the DTT consumption by SOA only, and $\Delta DTT_{imidazole}$ (pmol) is the DTT consumption by 2,4-dimethylimidazole only. As shown in Fig. 5, the enhancement factors of the four SOA were 2–3 orders of magnitude lower than those of the reference quinone compounds (i.e., NQN and PQN), suggesting that the redox cycling of quinones was not the major mechanism

underlying the DTT consumption by the SOA. Hamilton et al. (2005) reported that the total amount of identified quinones (i.e., 5-methyl-o-benzoquinone, 2-methyl-p-benzoquinone, 2-hydroxy-5-methyl-p-benzoquione) from the photooxidation of toluene was less than 0.07 % of the total aerosol mass. In a model compound study, Kumagai et al. (2002) also reported that the oxidation of DTT by most benzoquinones (e.g., 1,4-benzoquinone, 2-methyl-1,4-benzoquinone) was

negligible under simulated physiological conditions (pH = 7.5, 37 °C).



### 3.3 DTT modulator: non-catalytic particulate oxidizers

In-depth investigations on the roles of non-catalytic particulate oxidizers in DTT consumption were performed for isoprene SOA and toluene SOA, which led to high $DTT_t$. Organic hydroperoxides and PANs can oxidize sulfhydryl groups (oxidation state of $S[-2]$) to disulfides

($S[-1]$) or to even higher oxidation states ($S[0]$, $S[+2]$, $S[+4]$; Fig. 1b) (Grek et al., 2013). Under low $NO_x$ conditions, alkyl peroxy radicals ($RO_2$) dominantly react with $HO_2$ radicals, producing alcohols, alkyl hydroperoxides, and carbonyls (Finlayson-Pitts and Pitts Jr, 2000; Kroll et al., 2006; Ng et al., 2007b). Under high $NO_x$ conditions, $RO_2$ radicals mainly react with NO generating aldehydes (Finlayson-Pitts and Pitts Jr, 2000). The reaction of aldehydes with OH radicals

followed by the reaction with molecular oxygen yields peroxy acyl radicals ($RC(O)OO$) (Finlayson-Pitts and Pitts Jr, 2000). $RC(O)OO$ can react with $NO_2$ to form PANs and react with $HO_2$ radicals to form $RC(O)OOH$ (Finlayson-Pitts and Pitts Jr, 2000; Nguyen et al., 2012; Xu et al., 2014). In this study, the concentration of organic hydroperoxides per microgram of SOA were quantified using the NPBA assay, represented by $[OHP]_m$ (nmol $\mu g\text{-}SOA^{-1}$). The concentration of

PANs per microgram of SOA was measured by the Griess assay, represented by $[PAN]_m$ (nmol $\mu g\text{-}SOA^{-1}$).

As shown in Fig. 6a, by increasing the $HC/NO_x$ ratio from 5 to 27, $[OHP]_m$ in isoprene SOA increased 2 times owing to the organic hydroperoxides formed from the $RO_2+HO_2$ reaction pathway under low $NO_x$ conditions. Under the experimental conditions of this study, the influence

of $NO_x$ on $[OHP]_m$ in toluene SOA was insignificant. Presumably, the aging process reduced the significance of the $NO_x$ effect on $[OHP]_m$. Low-$NO_x$ toluene SOA was collected about 2 h later (i.e., a greater degree of aging) than high-$NO_x$ toluene SOA. The organic hydroperoxides in the low-$NO_x$ toluene experiment degraded more through photolysis or photooxidation (Lee et al., 2000) than those in the high-$NO_x$ toluene experiment. The effect of the aging process on toluene

$[OHP]_m$ was consistent with that on toluene $DTT_t$ (Fig. 3).

As shown in Fig. 6b, $[PAN]_m$ was found to be one order of magnitude lower than $[OHP]_m$. With the decrease in $HC/NO_x$ from about 22 to 9, $[PAN]_m$ in the toluene SOA increased 3 times as a result of PANs production from the $RO_2+NO$ reaction pathway under high $NO_x$ conditions (Fig. 1a) (Xu et al., 2014). For isoprene, the moles of both aerosol phase PANs and gas phase PANs per

cubic of air volume were significantly greater at higher $NO_x$ levels (Fig. S4). Most PAN products from the photooxidation of isoprene existed in the gas phase and the amount of PAN in particle





phase was trivial (Fig. S4); for example, aerosol phase PAN products was only 0.5 % of gas phase PAN products.

To investigate the effect of the interaction among different compounds on DTT response, the additivity of $DTT_m$ by various modulators was tested. As shown in Fig. S5, the $DTT_m$ of the

mixture of four model compounds (i.e., acrolein, PQN, $H_2O_2$, and tert-butyl hydroperoxides) was consistent with the sum of the $DTT_m$ originating from individual compounds, proving that $DTT_m$ is additive.

To underline the contribution of organic hydroperoxides and PANs to the $DTT_m$ of SOA, the $DTT_m$ values of toluene and isoprene SOA were also compared with the sum of $[OHP]_m$ and $[PAN]_m$.

Figure 6c shows that organic hydroperoxides were the major products that induced the oxidative potential of isoprene SOA. For toluene SOA, only 50–70 % of $DTT_m$ could be ascribed to organic hydroperoxides, and the remaining fraction was attributed to other organic compounds in SOA. We propose that electron-deficient alkenes, abundant in toluene SOA (Jang and Kamens, 2001), can substantially modify sulfhydryl groups in DTT via a Michael addition (Fig. 1b) (Nair et al.,

2014). In the next section, the reactivity of electron-deficient alkenes with DTT will be demonstrated using selected model compounds.

### 3.4 DTT modulator: electron-deficient alkenes

Figure 7 illustrates the $DTT_t$ ($t$ = 30 min) of four electron-deficient alkenes (i.e., acrolein, methacrolein, 2,4-hexadienal, and mesityl oxide). Acrolein showed much higher $DTT_t$ than the

other compounds. The susceptibility of an alkene to a Michael addition reaction depends on the nature of the electron withdrawing group coupled to the C=C bond (Nair et al., 2014). The methyl group of methacrolein and mesityl oxide is an electron donating group that increases the electron density on the C=C bond; thus, decreasing the reactivity of the C=C bond with DTT. The extended conjugation (C=C-C=C-C(O)H) in 2,4-hexadienal stabilizes the C=C bond leading to an extremely

low $DTT_t$.

The alkenes from the photooxidation of toluene were usually coupled with electron withdrawing groups such as carbonyls, nitrates, and carboxylic acids (Jang and Kamens, 2001). These electron-withdrawing groups enable the alkenes to be reactive with DTT. Compared with toluene SOA, TMB SOA will have more alkyl substituted alkenes owing to the three methyl groups on the

aromatic ring, and therefore be less reactive with DTT. This tendency partially explains why the



$DTT_t$ of TMB SOA was significantly lower than that of toluene SOA (Fig. 3). Based on aerosol composition predictions using predictive SOA models such as the Unified Partitioning Aerosol Phase Reaction (UNIPAR) model, the mass fraction of electron-deficient alkenes in high-$NO_x$ toluene SOA should be more than 50 % (Im et al., 2014); therefore, the gap between toluene $DTT_m$

and concentrations of non-catalytic particulate oxidizers (Fig. 6c) may be filled by abundant electron-deficient alkenes.

## 4 Atmospheric implications and conclusions

The influence of $NO_x$ on the oxidative potential of SOA was investigated using $DTT_t$ (Fig. 3). Among four HCs, only isoprene SOA was significantly sensitive to $NO_x$ levels, showing much

higher $DTT_t$ at lower $NO_x$ conditions. The $DTT_t$ of toluene SOA was found to be lower with a longer aging time, regardless of $NO_x$ conditions.

For SOA consisting of non-catalytic redox compounds, $DTT_m$ is more appropriate than $DTT_t$ for assessing oxidative potential, because of the non-linear relationship between DTT consumption and reaction time (Fig. 4). A decrease in isoprene $DTT_m$ was observed with increasing $NO_x$ levels,

but no significant $NO_x$ effect on $DTT_m$ was observed for toluene SOA within a 2h reaction. To apply the $DTT_m$ results of this study to ambient atmosphere, $DTT_m$ should be coupled with SOA mass concentrations. Under high $NO_x$ conditions, the $DTT_m$ of toluene SOA was almost 5 times higher than that of isoprene SOA, underlining the importance of toluene in urban areas, despite of its lower SOA yield (Table 1) in the urban environment (i.e., higher $NO_x$ conditions). In spite of

relatively low $DTT_m$ for high-$NO_x$ isoprene SOA, isoprene could still play a substantial role in the oxidative potential of ambient urban aerosols, because of its abundance (Guenther et al., 2006) and high SOA yields (Table 1) under high $NO_x$ conditions. The $NO_x$ effect on the $DTT_m$ of isoprene SOA is limited to the $NO_x$ conditions applied in this study, and should be extended to a variety of HC/$NO_x$ ratios in further studies.

As shown in Sect. 3.2 (DTT modulator: quinones), the DTT consumption by SOA was not sourced from quinones, which can catalytically yield ROS. Hence, the contribution of non-catalytic particulate oxidizers, especially organic hydroperoxides, to the oxidative potential of SOA was highlighted in this study. Non-catalytic particulate oxidizers account for almost 100 % of isoprene $DTT_m$, and 50–70 % of toluene $DTT_m$ (Fig. 6c). In addition to non-catalytic particulate oxidizers,





electron-deficient alkenes in toluene SOA can potentially react with DTT via a Michael addition (Nair et al., 2014).

The results of this study also show that some of the oxidizers (e.g., PANs) formed from the photooxidation of hydrocarbons predominantly exist in the gas phase (Fig. S4). Future studies should further consider how, through absorption into the bio-system, gas phase oxidizers may be effectual for inducing oxidative stress. Some products may be chemically unstable in aqueous solutions and decay during PILS sampling, inducing a negative artifact on the measurement of DTT consumption; therefore, the development of a cell-free assay without water-extraction of aerosols is needed to improve the assessment of the modulation capacity of SOA on cellular materials.

**Supplement.** The Supplement related to this article is available.

**Competing interests.** The authors declare that they have no conflict of interest.

**Acknowledgement.** This work was supported by a grant (2014M3C8A5032316) from the Ministry of Science, ICT, and Future Planning at South Korea.

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





Table 1. Outdoor chamber experiment conditions

| Date and Chamber | Initial HC ppb | Initial $NO_x$ (HONO) [a] ppb | $HC/NO_x$ ppbC/ppb | ΔHC ppb | $Y$ % | Mid-collection time [b] | RH [c] % | Temp [c] K | Chemical assay [d] |
|---|---|---|---|---|---|---|---|---|---|
| **Toluene** | | | | | | | | | |
| 13 Feb 2016 | 641 | 525 (193) | 9 | 403 | 15.1 | 13:40 | 22–63 | 281–303 | DTT |
| 13 Feb 2016 | 627 | 155 | 28 | 229 | 24.6 | 15:20 | 25–66 | 282–302 | DTT |
| 01 May 2016 | 935 | 766 (133) | 9 | 631 | 14.6 | 14:20 | 18–46 | 294–316 | DTT, PAN |
| 01 May 2016 | 938 | 301 (73) | 22 | 542 | 14.3 | 12:10 | 21–48 | 294–315 | DTT, PAN |
| 23 May 2016 | 691 | 906 (250) | 5 | 546 | 7.1 | 13:20 | 18–60 | 288–315 | DTT, Enhance |
| 23 May 2016 | 735 | 313 (86) | 16 | 421 | 9.3 | 15:40 | 15–22 | 307–316 | DTT, Enhance |
| 18 Aug 2016 | 640 | 783 (179) | 6 | 517 | 9.1 | 12:30 | 24–61 | 297–319 | DTT, OHP |
| 06 Aug 2016 | 610 | 240 (55) | 18 | 216 | 9.2 | 12:30 | 43–59 | 297–305 | DTT |
| 18 Aug 2016 | 342 | 107 (24) | 22 | 227 | 5.2 | 14:20 | 20–38 | 303–321 | OHP |
| 17 Nov 2016 | 622 | 179 (43) | 24 | 452 | 8.1 | 13:20 | 12–56 | 282–309 | DTT [e] |
| **TMB** | | | | | | | | | |
| 04 Oct 2015 | 613 | 920 | 6 | 613 | 6.7 | 14:40 | 20–43 | 290–310 | DTT |
| 04 Oct 2015 | 657 | 310 | 19 | 542 | 7.8 | 13:20 | 24–46 | 290–306 | DTT |
| 20 Feb 2016 | 589 | 1024 | 5 | 548 | 5.6 | 13:00 | 14–60 | 282–311 | DTT |
| 20 Feb 2016 | 583 | 156 | 34 | 455 | 5.7 | 14:40 | 16–61 | 282–311 | DTT |
| 11 Jan 2016 | 595 | 256 | 21 | 414 | 5.6 | 15:50 | 23–81 | 274–298 | Enhance |
| **Isoprene** | | | | | | | | | |
| 23 Apr 2016 | 2693 | 2680 | 5 | 2693 | 4.7 | 12:00 | 18–48 | 290–314 | DTT |
| 23 Apr 2016 | 2755 | 430 | 32 | 2755 | 1.2 | 13:30 | 23–51 | 290–312 | DTT |
| 14 May 2016 | 2928 | 2800 | 5 | 2928 | 5.0 | 14:20 | 17–47 | 292–315 | DTT, Enhance |
| 14 May 2016 | 2858 | 423 | 34 | 2858 | 1.3 | 12:00 | 25–55 | 293–312 | DTT |
| 22 Jul 2016 | 2525 | 2423 | 5 | 2525 | 3.5 | 13:20 | 20–55 | 297–320 | PAN (gas) [f] |
| 22 Jul 2016 | 2718 | 473 | 29 | 2718 | 0.9 | 12:50 | 23–58 | 297–320 | PAN (gas) [f] |
| 20 Aug 2016 | 3060 | 3300 | 5 | 3060 | 3.3 | 12:30 | 20–58 | 296–321 | DTT, OHP, PAN |
| 20 Aug 2016 | 3173 | 583 | 27 | 3173 | 1.4 | 11:50 | 25–61 | 297–318 | DTT, OHP, PAN |
| **α-Pinene** | | | | | | | | | |
| 25 Feb 2016 | 319 | 639 | 5 | 319 | 14.5 | 15:00 | 21–63 | 278–299 | DTT |
| 25 Feb 2016 | 323 | 91 | 36 | 323 | 36.1 | 13:30 | 25–67 | 278–298 | DTT |
| 18 Jan 2016 | 257 | 144 | 18 | 257 | 15.6 | 15:50 | 25–78 | 275–297 | Enhance |

[a] For toluene experiments, $NO_x$ was contributed by NO, $NO_2$ and HONO. The concentration of HONO was estimated using the difference in the $NO_2$ signal with and without the base denuder (1 % $Na_2CO_3$+1 % glucose).

[b] This column is the mid-collection time of SOA sampling.

5 [c] The RH and temperature conditions shown in the Table 1 were recorded from the beginning of photooxidation (sunrise) until the ending of PILS sampling.

[d] The SOA samples were applied to a series of chemical assays, namely DTT assay (DTT), DTT enhancement (Enhance), organic hydroperoxides analysis (OHP), and PAN analysis (PAN)

[e] For DTT measurement of toluene SOA sample collected on 17 Nov. 2016, the concentration of potassium phosphate

10 buffer (0.8 mM) in the first step of DTT assay was two times higher than the typical buffer concentration (0.4 mM). The $DTT_m$ of the toluene SOA sample (17 Nov. 2016) is shown in Fig. 4.

[f] The concentration of gaseous PAN products (collected by an impinger) was measured by the Griess assay.




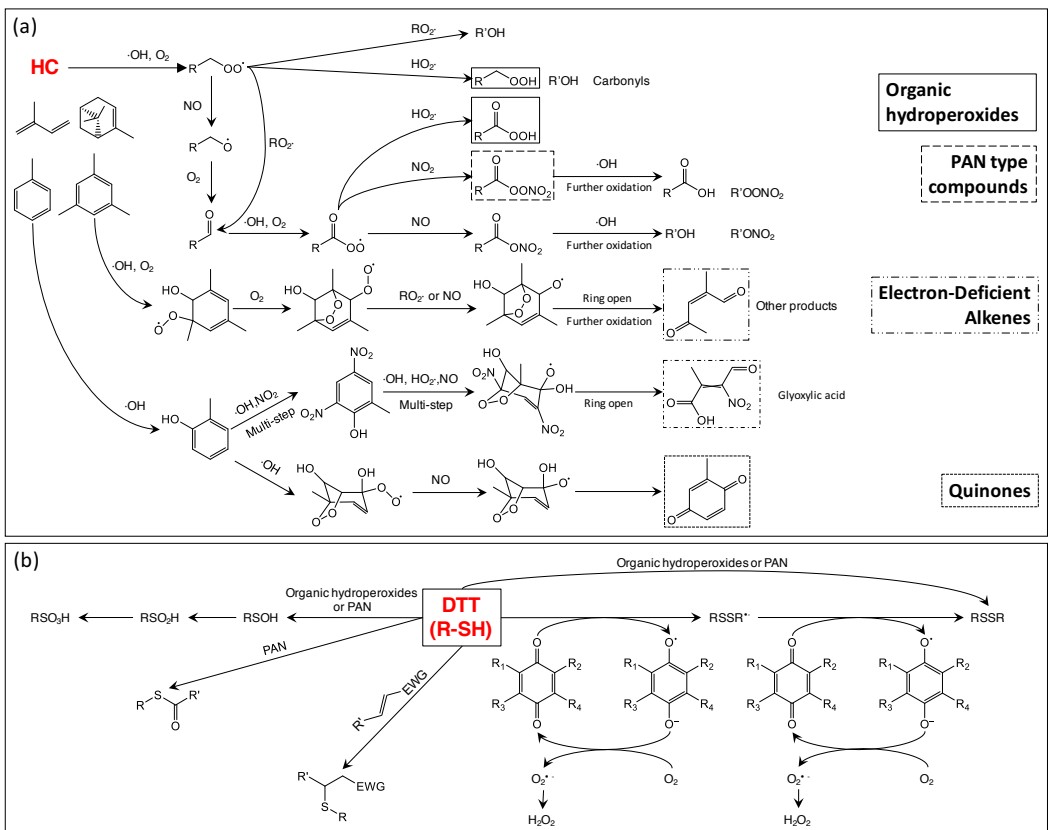

**Figure 1. (a) Simplified mechanisms for the formation of alkyl and acyl hydroperoxides, peroxy acyl nitrates (PANs), electron-deficient alkenes, and quinones (Eddingsaas et al., 2012b; Jang and Kamens, 2001; Saunders et al., 2003, 1997; Wyche et al., 2009; Xu et al., 2014). Photooxidation products are not limited to the compounds shown. (b) Possible reaction mechanisms between sulfhydryl groups in dithiothreitol (DTT, represented by R-SH) and SOA products (Grek et al., 2013; Kumagai et al., 2002; Mudd, 1966; Nair et al., 2014). EWG represents the electron-withdrawing group attached to an alkene.**





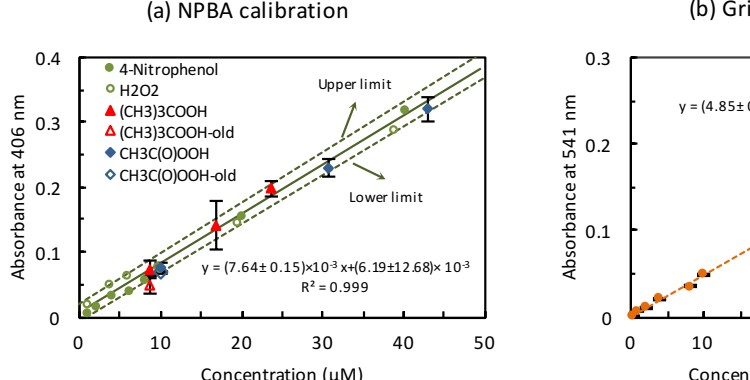

**Figure 2. (a) Calibration curve of 4-nitrophenylboronic acid (NPBA) assay obtained at the 90 % confidence level. The calibration curve was applied to test the feasibility of using NPBA assay to quantify the concentration of $H_2O_2$, tert-butyl hydroperoxides (($CH_3$)$_3$COOH), and peracetic acid ($CH_3$C(O)OOH). The ($CH_3$)$_3$COOH-old and $CH_3$C(O)OOH-old represent the chemicals that were stored in a refrigerator at 4 °C for one week. (b) Calibration curve of Griess assay. The error bar associated with each data point was calculated by $t_{0.90} \times \sigma/\sqrt{n}$, where $t_{0.90}$ is the t-score (2.920 for $n = 3$ replicates) with a two-tail 90 % confidence level, and $\sigma$ is the standard deviation.**





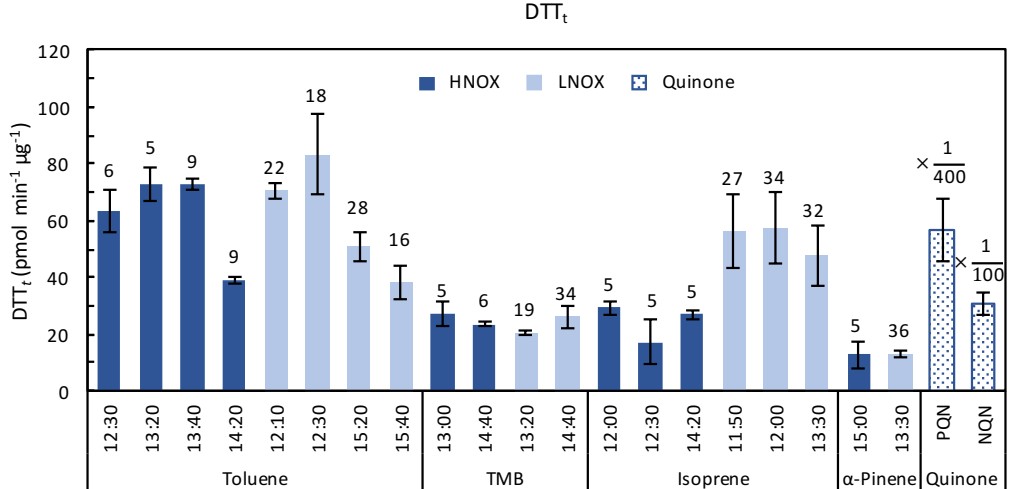

**Figure 3.** $DTT_t$ **of chamber generated SOA under varied** $NO_x$ **conditions (HNOX: high** $NO_x$**, LNOX: low** $NO_x$**) and positive controls (i.e., PQN and NQN). The number above each column represents the** $HC/NO_x$ **ratio. The x-axis represents the mid-collection time (Table 1). The** $DTT_t$ **of PQN and NQN are divided by 400 and 100, respectively. Each error bar was calculated by** $t_{0.95} \times \sigma/\sqrt{n}$**, where** $t_{0.95}$ **is the t-score (4.303 for** $n = 3$ **replicates) with a two-tail 95 % confidence level.**



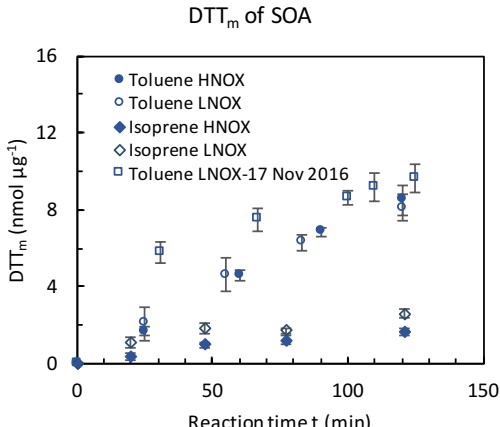

**Figure 4. Time profile of $DTT_m$ for toluene and isoprene SOA under different $NO_x$ conditions (HNOX: high $NO_x$, LNOX: low $NO_x$). Toluene LNOX-17 Nov 2016 represents the toluene sample (collected on 17 Nov. 2016) tested with a 0.8 mM potassium phosphate buffer in the first step of DTT assay (2 times higher than the typical buffer concentration (0.4 mM)). Each error bar was calculated by $t_{0.95} \times \sigma / \sqrt{n}$ using three replicates, where $t_{0.95}$ is the t-score (4.303 for $n = 3$ replicates) with a two-tail 95 % confidence level.**

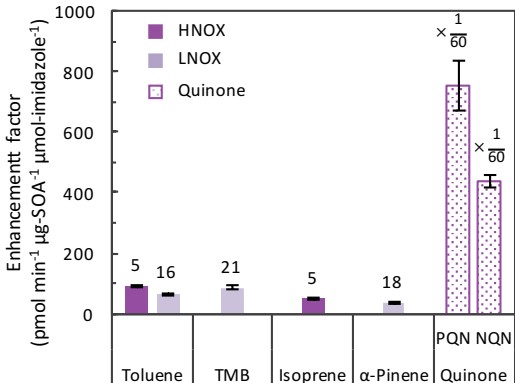

**Figure 5. Enhancement factors (pmol min$^{-1}$ μg-SOA$^{-1}$ μmol-imidazole$^{-1}$) of SOA in the presence of 2,4-dimethylimidazole (HNOX: high $NO_x$, LNOX: low $NO_x$). The label above each column represents the HC/$NO_x$ ratio. The enhancement factor is expressed as the mean (± σ) of three replicates. The enhancement factors of PQN and NQN are divided by 60.**



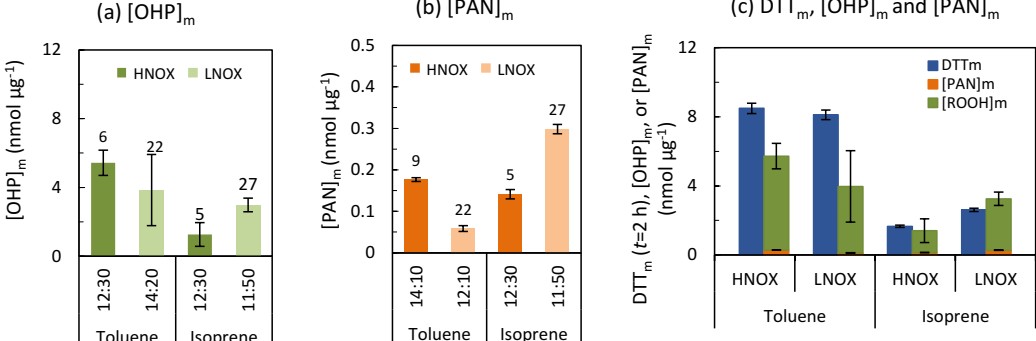

**Figure 6. (a) Concentration of organic hydroperoxides in SOA, $[OHP]_m$ (nmol $\mu g^{-1}$), measured by 4-nitrophenylboronic acid (NPBA) assay. (b) Concentration of PANs in SOA, $[PAN]_m$ (nmol $\mu g^{-1}$), measured by Griess assay. The number above each column represents the HC/NO$_x$ ratio. The x-axis represents the mid-collection time (Table 1). (c) Comparison of $DTT_m$ (t = 2 h) with the sum of $[OHP]_m$ and $[PAN]_m$. The $[OHP]_m$, $[PAN]_m$, and $DTT_m$ are expressed as the mean (± σ) of three replicates. HNOX represents high NO$_x$ conditions, and LNOX represents low NO$_x$ conditions.**

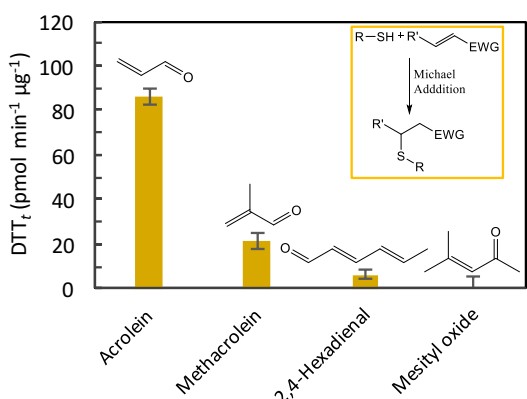

**Figure 7. The $DTT_t$ (t = 30 min) of four different electron-deficient alkenes. Each error bar was calculated by $t_{0.95} \times \sigma / \sqrt{n}$ using three replicates, where $t_{0.95}$ is the t-score (4.303 for n = 3 replicates) with a two-tail 95 % confidence level. EWG appeared in the mechanism represents an electron withdrawing group (Nair et al., 2014).**