# Peer review of "Dithiothreitol Activity by Particulate Oxidizers of SOA Produced from Photooxidation of Hydrocarbons under Varied $NO_x$ Levels"

_Atmospheric Chemistry and Physics, 2017_

## Referee Comment (RC1) · Anonymous Referee #1 · 16 Apr 2017

General comments: This manuscript presents a laboratory chamber study to investigate the contribution of particulate oxidizers in secondary organic aerosols to the oxidative potential measured dithiothreitol (DTT) assays. The significance of quinones, organic hydroperoxides, and peroxyacyl nitrates (PANs) in SOA samples from various precursors and conditions was characterized, and directly compared to the measured DTT activity. The authors report that the presence of particulate organic hydroperoxides can explain most of the DTT consumption of isoprene SOA (almost 1:1 correlation), and partially for toluene SOA, while the amount of PANs and quinones appears to be negligible in the SOA systems investigated here. With completely different approaches, it is intriguing to see the results reported in this study to be closely consistent with Kramer et al. (Atmos Environ, 2016) who directly measured the DTT activity of synthesized isoprene-derived hydroperoxide (ISOPOOH) standard and reported a very strong DTT response (almost comparable to the quinone compound and 3 order of magnitudes higher than other major isoprene SOA constituents).

Overall, this is a well-planned study and the manuscript is well written. I have a few specific questions and suggestions are listed below for the authors' clarification and consideration. I am in support of publication once these questions have been addressed.

Specific comments: Page 5, Line 24: should the storing condition here be 4 °C instead of −4 °C? Page 6 Line 2: how the reaction between DTT and SOA could be quenched by adding 1 mL trichloroacetic acid (1 % w/v)? Was this to quench specific components in SOA samples before measuring the remaining DTT with DTNB? More detailed explanation is needed as it is not clear here. Page 6 Line 13-17: how the slopes shown in Figure S1 compare to each other statistically? Table 1. It looks like no seed aerosols were introduced into the chamber experiments. Did the SOA form solely because of nucleation? How the chemical composition and DTT activity might change if there are pre-existing seed aerosols? Figure 4: why a higher concentration of potassium phosphate buffer was needed for the Toluene LNOX-17 Nov 2016 sample? Would changing buffer concentrations affect the measured DTT activity?
* * *

---

## Referee Comment (RC2) · Anonymous Referee #2 · 17 Apr 2017

In this work, Jiang et al. studied the oxidative potential of Secondary Organic Aerosol (SOA) generated in laboratory chambers. The authors investigated the role of various oxygenated compounds within complex SOA systems in contributing to oxidative decay of sulfhydryl groups, which is believed to be the main mechanism by which particulate matter causes adverse cardiopulmonary outcomes. The authors systematically varied experimental conditions and carefully measured various functional groups, and relate the composition to OP measured (using the DTT assay). Based on the experimental results, they conclude that organic hydroperoxides (ROOH) are a major source of oxidative potential, and suggest that electron-deficient alkenes (such as acrolein) may also be important too.

The experiments are done with sufficient controls, and carefully interpreted. The DTT activity of SOA has recently become a topic of great interest within the atmospheric community. The work is within the scope of ACP and should be published after considering the following minor comments.

Major comments:

- The suggestion about electron-deficient alkenes being important is worth noting. Acrolein seems to be the only example that has significant DTT activity. All others are an order of magnitude lower. Are there example compounds that are more relevant to toluene/benzene system? I would imagine that aromatic compounds would yield conjugated compounds, which according to Figure 7 and Section 3.4, would have negligible DTT activity. Also, acrolein has very high vapor pressure and is unlikely to be in the particle phase.

- Related to the previous note, in Section 3.1, the decrease in DTT activity is on the same timescale of acrolein. That is not a fair comparison because acrolein is reacting in the gas phase, where the DTT measurement is for compounds in the particle phase, which likely have longer lifetimes than acrolein in the gas phase.

- On that note, why is it called "electron-deficient alkenes"? Why not "unsaturated carbonyls"? Reaction with DTT is likely on the oxygenated group, rather than on the C=C double bond. Also, from a fundamental perspective, does the C=C double bond need to adjacent to the C=O group? Have the authors investigated, say, 3-pentenal?

- Also, if acrolein indeed contributes high DTT activity, I would imagine that SOA from 1,3-butadiene would have very high DTT activity. Have the authors done any experiments to suggest that is the case?

- Another suggestion for additional experiments (optional but would really strengthen the argument): if the authors can pass ozone over the SOA (e.g. collected on the filter), the ozone should selectively remove the double bonds. This way the authors

can isolate the contribution to DTT from these electron deficient alkenes. While this may create additional organic hydroperoxides, the authors can easily correct for that increase by measuring total OHP with the NPBA assay.

- Does the UNIPAR model describe the abundance of electron deficient alkenes under low-NOx? I would imagine under high NOx there would be more electron deficient alkenes where C-C bond scission and ring-opening reactions are dominant, but the "unaccounted" DTT activity seems to be higher under low NOx for toluene (from Fig. 6).

- I understand that the proposal about measuring DTTm as a proxy for maximum capacity (described in Section 3.1) is useful for a chemist. But I wonder what the biological relevance is. What is the lifetime of particles deposited in the lungs? I am not a lung expert, but I wonder if mucociliary clearance would make the lifetime shorter than 2 hours, and therefore DTTm at 2 h may not be that relevant.

- For DTTm to be meaningful, DTT must be in large excess compared to amount of SOA. What is the estimated ratio of DTT to SOA (100 uL of DTT, 700 uL of SOA solution)? Is DTT in large excess? This is not an issue when looking at the catalytic activity, because both DTT and quinones can be regenerated through catalytic cycles. But this issue arised when it is a one-step reaction with these particulate oxidizers.

- If hydroperoxides (and PANs) are indeed unstable at room temperature (as suggested in Section 2.3.2), did the decrease in ROOH amount correspond to an increase in DTTt or DTTm?

- There are large fluctuations in experiment temperatures (more than 20K). Are there systematic differences in chemical composition?

Minor comments:

- Section 2.2: Please provide define PILS and provide details of the PILS method. Was steam used to grow particles for impaction and collection? If so, is there concern

about the high temperatures leading to decomposition of thermally labile compounds important for oxidative potential?

- Section 2.3.1 Line 1: it is unclear what "two steps" mean in parentheses. Perhaps re-word to "The two-step DTT assay..."

- Section 2.3.1 Line 2: What does SOA sample mean? Is this the aqueous solution?

- Section 2.3.1 Page 6 Line 1: What is the pH of the PBS?

- Section 2.3.1 Page 6 Line 1: "shaken in a sonicator" is confusing. Particularly since in the later section "shaking" was compared to "sonicating".

- Section 2.3.2: Can 4-nitrophenol be formed in high NOx SOA from toluene? If so, are there any negative controls?

- Section 3.1 heading: remove period after title

- Section 3.3 Line 5: Grek et al. does not provide any evidence that PANs can oxidize sulfhydryl groups. They only include peroxynitrites as an RNS, but PANs are peroxy nitrates.

- Section 4 Line 25: remove (DTT modulator: quinones)

- Section 4 Page 14 Line 8: An assay that is not water-based may not be relevant, since cells are water-based

- Table 1 and throughout manuscript: Does HC/NOx ratio refer to the initial or average HC/NOx? If it is the average, over what time range is the HC/NOx calculated?

- Table 1 Mid-collection time: is this local time?

- Figure 4: why is one data series labeled with experiment date, but others are not?

---

## Referee Comment (RC3) · Anonymous Referee #3 · 5 May 2017

This study investigated the DTT activity of different types of SOA (toluene, TMB, isoprene, a-pinene) formed in the presence of NOx. Experiments are conducted in an outdoor chamber facility with different levels of NOx. Results showed that in the presence of relatively higher levels of NOx, DTT activity of toluene SOA is 2-5 times higher than other SOA. Isoprene SOA has lower DTT with increasing NOx. Other SOA appears to be insensitive to NOx. The results are discussed in the context of different DTT modulator compounds.

This is an interesting study and will be of interest to the greater research community. I have a number of questions regarding the experimental design/protocols and how the data are interpreted. Further, the conclusions need to be better justified.

[Figure]

The experiments are conducted with very high levels of hydrocarbon and NOx. It is not clear what the NOx levels in this study mean in terms of the reaction pathways. It appears that the authors assume that under the "low-NOx" conditions in this study, the dominate fate of the peroxy is reaction with HO2. It is not immediately clear how this is the case, especially with the high levels of hydrocarbon and NOx, and without an external HO2 source. To what extent does RO2+HO2 proceed in these experiments and whether organic hydroperoxide should be produced in the first place?

The results in this study should be compared with prior literature and discussed. For instance, Kramer et al. (2016) found that for isoprene SOA, high-NOx conditions produce SOA that is more oxidizing compared low-NOx conditions. But, the results in this work suggest the opposite. Results from SOA formed from different hydrocarbons should also be compared to prior literature when available (e.g., Rattanavaraha et al., 2011, Tuet et al., 2017a; Tuet et al., 2017b). Tuet et al. found the DTT activity of different SOA to be fairly insensitive to RO2 fate.

Further, the same authors of this manuscript published a recent paper in Atmos Environ (Jiang et al. 2016) on similar experiments, but the results under similar reaction conditions are not the same as those reported in this study. Please clarify and discuss accordingly.

It was noted that the SOA yields in this study are consistent with prior studies. SOA yields should be compared in the context of organic mass loadings. It does not appear that the yields in this work are in line with literature. Based on the numbers provided in Table 1, the SOA yields calculated are very different from prior studies. Please see detailed comments below. The implications of these differences should be considered and discussed. If the yields are so different, what does it imply regarding their corresponding SOA composition and their health effects? On a related note, the organic mass loadings (one can calculate them based on Table 1) are high in these experiments, which will result in partitioning of more volatile species into the particle phase compared to ambient. The implications of this on the measured DTT activity in this
study and how they can be applied to ambient should also be discussed.

The authors attributed the difference in [OHP]m between low and high NOx experiments to sample collection times and associated aging. I do not think this is well-justified. Many other parameters are also changing at the same time.

Finally, the basis for comparison of DTT activity is time of the sample collection in many cases. Is that chosen to represent OH exposure? But if so, it is not clear that [OH] are constant and comparable across different experiments.

Overall, I recommend publication with major revisions. More detailed comments below.

1. Page 1, line 14. Define clearly what is considered as high vs. low NOx conditions in this study.

2. Page 4, methods.

a. Was seed aerosols used in the experiments? Please state clearly.

b. The NOx levels in this study were very high in all experiments. The authors need to define LNOx and HNOx clearly in the methods.

c. HONO was used only in toluene experiments, why? The other experiments used NOx (was that NO or NO2? Or both? In what ratio?)

d. The authors should provide figures (in the SI) to show the typical time series of hydrocarbon, NO, NO2, ozone, SOA mass for an experiment using HONO vs an experiment using NOx. This is critical to give some context regarding the conditions under which the SOA samples were collected for DTT analysis. (for instance, what was the NOx levels when the SOA samples were collected? This has important implications on the SOA composition and whether the results from different SOA samples can be directly compared.)

e. What is the collection efficiency of the PILS? How is it calculated? How is it validated? As the DTT values was normalized by mass, the authors need to provide more

details to justify the accuracy of the SOA mass values used in the DTT calculations.

3. Page 6, line 17. It was noted that "The SOA mass applied to DTT assay was constrained to ensure that the DTT consumption remained less than 50 % of DTT0." Why?

4. Page 6, line 20 onwards, the NPBA assay. It is not clear how the authors treat (or correct for) potential interference from other SOA components in the NPBA assay measurements.

a. It was noted that absorbance at 406 nm (authors referring to 4-nitrophenol) of SOA before reaction with NPBA was negligible. I do not understand why this is the case. The oxidation of aromatic compounds (such as toluene) by OH in the presence can result in the formation of 4-nitrophenol? If so, why is the absorbance at 406nm negligible?

b. In terms of potential interference from alcohols, it was noted that glycerol aqueous solutions were tested using the NPBA method and no measurable absorption appeared in UV spectrum. The authors then concluded that the UV absorption spectrum was originated from 4-nitrophenol. Multi-functional alcohols can be formed from the oxidation of hydrocarbons. How do the authors justify that the results from glycerol is representative and that the measured absorption is purely from 4-nitrophenol formed by NPBA reactions with organic peroxides, without any interference from other SOA components?

c. Why "no more than 10 ug SOA was applied to the NPBA assay"?

5. Page 7, line 16 onwards, PAN analysis. Please provide further details to show that PAN are completely hydrolyzed in 15 mins.

6. Page 8, line 14-18. SOA yields from different hydrocarbons cannot be compared without the context of organic aerosol mass, as yield is typically a function of organic aerosol mass (Odum equation).

a. In Table 1, while one can calculate the organic mass loading from deltaHC and

yield, the authors should also provide the organic mass loadings in the table to facilitate easier comparison of yields from this study to prior studies.

b. I quickly did the calculations, but found that the yields for the different hydrocarbons are very different from the previous studies cited in the manuscript. For instance, the yield for isoprene SOA photooxidation (with NOx/isoprene ratio of ~5) is all ~ 5% for Kroll et al., Xu et al., and this study. However, the organic mass loading that corresponds to this 5% SOA yield in Kroll et al and Xu et al is ~ an order of magnitude smaller than this study. This means that when plotted in the Y vs. deltaMo (Odum equation) space, the SOA yields under this specific NOx/isoprene ratio in this study is substantially lower than all previous studies. Discrepancies also exist for other hydrocarbons. The authors should conduct a detailed comparison with previous studies by showing their data in the yield curve space and comparing with others. The discrepancies should be discussed.

7. Page 9, line 8-15. Comparison of response of aerosols from different hydrocarbons. This section needs to be expanded to include more discussions (in addition to the description of results).

a. The authors noted that a previous study by Fujitani et al. (2012) with epithelial cells is consistent with this study. Can the cellular results from Fujitani et al. (2012) be directly compared to the DTT in this study? Please discuss.

b. How do results in this work compare to previous studies? For instance, recent work from Tuet et al. showed that the response from naphthalene is higher than other hydrocarbons such as m-xylene and a-pinene; McWhinney has also demonstrated previously that showed naphthalene SOA is highly redox-active. It would be useful that the authors provide some context and discuss the DTT activity of different SOA with respect to previous work.

8. Page 9, line 25. It was stated that "...DTTm of low-NOx isoprene SOA was much higher than that of high-NOx isoprene SOA". This does not seem like it is the case from

Figure 4. Some data points overlap and are within uncertainty.

9. Page 9, line 27. (and page 10) But for the "toluene LNOx-17 Nov 2016", the data are not linear? i.e., will the data also start from the origin, if so, considering the origin and the 5 data points in Figure 4, the overall trend is then non-linear? Please discuss.

10. Page 10, quinones. The authors stated that "the redox cycling of quinones was not the major mechanism underlying DTT consumption by the SOA". A small contribution of quinones to the total aerosol mass does not necessarily mean they are not important for overall toxicity? Oxidation of the aromatic compounds used in this study can lead to formation of quinones (e.g., Bloss et al., 2005). Many previous studies have pointed to the importance of quinones in ambient PM toxicity. It is not clear how the results in this work should be placed in the context of previous work that pointed to the importance of quinones for PM toxicity. Please discuss.

11. Page 11, NOx conditions.

a. Line 5 onwards. The authors noted that under low NOx conditions, RO2 predominately reacts with HO2, producing hydroperoxides (among other products). This accuracy of this statement will depends on what precisely the "low NOx conditions" are, as "low NOx conditions" do not directly (or necessarily) translate to RO2+HO2 reactions. It is not clear that RO2+HO2 is the dominant reaction under the conditions of this study. The hydrocarbon concentration used in this study is very high (hundreds of ppb to several ppm), there is abundant NOx (even under "low NOx" conditions), but no addition HO2 source (such as H2O2). With this, it is not clear how RO2+HO2 dominates. Since a large fraction of the discussions in the manuscript hinged on this, it is critical that this is justified clearly in the manuscript. The authors can perform a simple simulation of the relative importance of different RO2 reactions under their "low" and "high" NOx conditions.

b. The authors shall compare and discuss their results in the context of previous studies. For instance, a recent Kramer et al. (2016) concluded that High-NOx con-

ditions produce SOA that is more oxidizing compared low-NOx conditions. This re-sults from this work showed the opposite. Please discuss. Also, Tuet et al. (2017a, 2017b) specifically studied the toxicity of SOA (including isoprene) under RO2+HO2 vs RO2+NO reactions conditions, and found DTT activity of isoprene SOA to be similar under RO2+HO2 and RO2+NO conditions, also for other SOA except naphthalene.

c. How do the data in this study compare the results from a previous study by the same author (Jiang et al. AE, 2016). For example, the "without denuder" data in Figure 2 Jiang et al AE paper can be compared to those in the current study. But comparing this study and their previous AE publication, the results (in terms of the DTTt values) are different for each hydrocarbon under similar conditions? Please compare and discuss, and confirm self-consistency if that is the case.

d. Line 20, and figure 6. What is the x-axis (time) supposed to be a surrogate of? If it is supposed to be a surrogate for OH exposure, then the OH level should be the same is each experiment. Is this the case? The authors simply explained the differ-ence in [OHP]m between low and high NOx toluene SOA as the low-NOx SOA being collected at a later time and resulted in a lower level of [OHP]m due to further reac-tions/photooxidation. I do not think the authors can discount other factors, such as the varying RH (maybe SOA composition is different due to different RH?), organic mass loading (when I used the numbers in Table 1 and calculated the mass loadings for the toluene experiments shown in Figure 6, the loadings are very different for the two experiments), etc?

12. Table 1. Was ozone present in these experiments? If so, please include some in-formation here. This should also be specified and discussed clearly in the manuscript, in case some of the SOA is formed from ozonolysis in addition to OH oxidation.

Minor Comments:

1. Page 1, line 26. Clearly state in the abstract under what NOx conditions the sen-tence "The amount of organic hydroperoxide was substantial, while PANs were found

to be insignificant for both SOA." 2. Page 1, line 29. Clearly state what "model com-pound study" refers to. 3. Page 2, line 8. The author should also cite McDonald et al. (2012, Inhal. Toxicol), McWhinney et al. (2013, ACP), Kramer et al., (AE, 2016), Tuet et al. (2017, ACP), and Tuet et al. (2017, ACPD). 4. Page 4, line 7-9. The authors should state clearly that only selected (but not all) toluene and isoprene SOA samples are analyzed with the Griess and NPBA assays. 5. Page 5, line 16. Was 23 May 2016 a typo? Should it be July 22? (based on Table 1)

---

## Author Comment (AC1) · 3 Jun 2017

Please find the response in the supplement.

Please also note the supplement to this comment:
http://www.atmos-chem-phys-discuss.net/acp-2017-214/acp-2017-214-AC1-supplement.pdf

---

## Author Comment (AC2) · 3 Jun 2017

Please find the response in the supplement.

Please also note the supplement to this comment:
http://www.atmos-chem-phys-discuss.net/acp-2017-214/acp-2017-214-AC2-supplement.pdf

---

## Author Comment (AC3) · 3 Jun 2017

Please find the response in the supplement.

Please also note the supplement to this comment:
http://www.atmos-chem-phys-discuss.net/acp-2017-214/acp-2017-214-AC3-supplement.pdf

---

## Author Response (AR1)

**Response to comments from referee #1 (Manuscript Ref. NO: acp-2017-214)**

Huanhuan Jiang et al.
mjang@ufl.edu

Thank you for your valuable comments. We modified the manuscript accordingly.

**General comments:** This manuscript presents a laboratory chamber study to investigate the contribution of particulate oxidizers in secondary organic aerosols to the oxidative potential measured dithiothreitol (DTT) assays. The significance of quinones, organic hydroperoxides, and peroxyacyl nitrates (PANs) in SOA samples from various precursors and conditions was characterized, and directly compared to the measured DTT activity. The authors report that the presence of particulate organic hydroperoxides can explain most of the DTT consumption of isoprene SOA (almost 1:1 correlation), and partially for toluene SOA, while the amount of PANs and quinones appears to be negligible in the SOA systems investigated here. With completely different approaches, it is intriguing to see the results reported in this study to be closely consistent with Kramer et al. (Atmos Environ, 2016) who directly measured the DTT activity of synthesized isoprene-derived hydroperoxide (ISOPOOH) standard and reported a very strong DTT response (almost comparable to the quinone compound and 3 order of magnitudes higher than other major isoprene SOA constituents).

Overall, this is a well-planned study and the manuscript is well written. I have a few specific questions and suggestions are listed below for the authors' clarification and consideration. I am in support of publication once these questions have been addressed.

**Specific comments:**
**1.** Page 5, Line 24: should the storing condition here be 4 °C instead of −4 °C?

**Response:** This sentence was modified and reads now,
"Before chemical analysis, all SOA samples were stored in a refrigerator at 4 °C."

**2.** Page 6 Line 2: how the reaction between DTT and SOA could be quenched by adding 1 mL trichloroacetic acid (1 % w/v)? Was this to quench specific components in SOA samples before measuring the remaining DTT with DTNB? More detailed explanation is needed as it is not clear here.

**Response:** Trichloroacetic acid is a strong organic acid (pKa = 0.5) and commonly used as a quencher of thiol oxidation (Cho et al,. 2005, Fang et al. 2014, Curbo et al. 2013) by decreasing pH. Trichloroacetic acid is crystal. It is easy to prepare the quenching solution by weighing trichloroacetic acid. This sentence was modified as below:
"…the reaction between DTT and SOA was quenched by adding 1 mL 1% w/v trichloroacetic acid (a commonly used quencher of thiol oxidation)."

Reference: Cho, et al., Redox activity of airborne particulate matter at different sites in 30 the Los Angeles Basin, Environ. Res., 99, 40-47, 2005.
Sophie Curbo et al., Is Trichloroacetic Acid an Insufficient Sample Quencher of Redox Reactions? Antioxidants & Redox Signaling, 18, 2013.

Ting Fang et al., A semi-automated system for quantifying the oxidative potential of ambient particles in aqueous extracts using the dithiothreitol (DTT) assay: results from the Southeastern Center for Air Pollution and Epidemiology (SCAPE), Atmospheric Measurement Techniques, 2014

3. Page 6 Line 13-17: how the slopes shown in Figure S1 compare to each other statistically?

**Response:** With significance level $\alpha = 0.05$, the slope of $\Delta$DTT vs. time with sonicator was not significantly different from the one with shaker, tested using the statistical method which is based on the Student t-test (Andrade and Estévez-Pérez, 2014)). This sentence was modified and reads now,
"Tested using the statistical method based on the Student t-test (Andrade and Estévez-Pérez, 2014), the DTT loss rate in blank control during sonication was not significantly different from that with shaker (significance level $\alpha = 0.05$); therefore, in this study, the influence of free radicals generated by sonication on DTT measurement was insignificant."

**4.** Table 1. It looks like no seed aerosols were introduced into the chamber experiments. Did the SOA form solely because of nucleation?

**Response:** No, SOA was formed without seed aerosol.

**5.** How the chemical composition and DTT activity might change if there are pre-existing seed aerosols?

**Response:** It is a valuable question to think. With pre-existing aqueous seed aerosols, SOA yields will increase and SOA compositions will also change. We cannot simply answer for this question because aerosol formation is complex. What we can think is that the water content and aerosol acidity can influence the lifetime of unstable compounds, partitioning of organic species, and the reaction with atmospheric oxidants in particle phase.

**6.** Figure 4: why a higher concentration of potassium phosphate buffer was needed for the Toluene LNOX-17 Nov 2016 sample? Would changing buffer concentrations affect the measured DTT activity?

**Response:** Please find the last paragraph of Section 3.1.

"Medina-Ramos et al. (2013) reported that the electron transfer rate between glutathione (GSH) and an electrogenerated mediator ($[IrCl6]^{2-}$) exhibited a slight acceleration when the phosphate buffer concentration was increased from 0 to 50 mM at pH = 7.0. To achieve the completion of the reaction between particle oxidizers in SOA and DTT, the $DTT_m$ of toluene SOA (HC/NO$_x$=24 ppbC/ppb) was measured with a 0.8 mM potassium phosphate buffer in the first step of DTT assay (2 times higher than the typical buffer concentration (0.4 mM))."

Thus, the higher concentration of potassium phosphate buffer was applied to toluene (17 Nov 2016) sample and accelerated the rate of the reaction between DTT and toluene SOA.

**Response to comments from referee #2 (Manuscript Ref. NO: acp-2017-214)**

Huanhuan Jiang et al.
mjang@ufl.edu

Thank you very much for the thoughtful and constructive comments. The paper was greatly improved based on your suggestions.

In this work, Jiang et al. studied the oxidative potential of Secondary Organic Aerosol (SOA) generated in laboratory chambers. The authors investigated the role of various oxygenated compounds within complex SOA systems in contributing to oxidative decay of sulfhydryl groups, which is believed to be the main mechanism by which particulate matter causes adverse cardiopulmonary outcomes. The authors systematically varied experimental conditions and carefully measured various functional groups, and relate the composition to OP measured (using the DTT assay). Based on the experimental results, they conclude that organic hydroperoxides (ROOH) are a major source of oxidative potential, and suggest that electron-deficient alkenes (such as acrolein) may also be important too.
The experiments are done with sufficient controls, and carefully interpreted. The DTT activity of SOA has recently become a topic of great interest within the atmospheric community. The work is within the scope of ACP and should be published after considering the following minor comments.

**Major comments:**
**1.** The suggestion about electron-deficient alkenes being important is worth noting. Acrolein seems to be the only example that has significant DTT activity. All others are an order of magnitude lower. Are there example compounds that are more relevant to toluene/benzene system? I would imagine that aromatic compounds would yield conjugated compounds, which according to Figure 7 and Section 3.4, would have negligible DTT activity. Also, acrolein has very high vapor pressure and is unlikely to be in the particle phase.

**Response:** Acrolein, methyl acrolein, 2,4-hexadienal and mesityl oxide were chosen as model compounds to show how electron-deficient alkenes increase DTT activity. The DTT activity of alkenes increases with an electron withdrawing group such as carbonyls but decreases with an electron donating group such as alkyl groups. The electron-deficiency of acrolein is relatively higher than methyl acrolein, 2,4-hexadienal and mesityl oxide (conjugated carbonyl with 2 methyl groups). As discussed in Section 3.4, the toluene SOA contains a large amount of alkene that are coupled with electron withdrawing groups such as carbonyls, nitrates, and carboxylic acids. These electron-deficient alkenes in toluene SOA might contribute to the DTT response.

**2.** Related to the previous note, in Section 3.1, the decrease in DTT activity is on the same timescale of acrolein. That is not a fair comparison because acrolein is reacting in the gas phase, where the DTT measurement is for compounds in the particle phase, which likely have longer lifetimes than acrolein in the gas phase.

**Response:** In order to respond to the reviewer, the lifetime of two semi-volatile conjugated carbonyls (4-oxo-2-butenoic acid, and 2-hydroxy-3-penten-1,5-dial) that were reported in the previous study (Jang et al., 2001) was estimated using a structure-reactivity relationship for the reaction with OH radicals (chamber condition). The calculation was included in SI (Section S5). If these two electron-deficient carbonyls are oxidized by OH radicals in the gas phase, the estimated lifetime of 4-oxo-2-butenoic acid and 2-hydroxy-3-penten-1,5-dial is estimated to be 134 min and 43 min, respectively. The actual lifetime of these compounds will be shorter than our estimation since they can also be oxidized in particle phase.

Reference: Jang, M. and Kamens, R. M.: Characterization of secondary aerosol from the photooxidation of toluene in the presence of NOx and 1-propene, Environ. Sci. Technol., 35, 3626-3639, 2001.

**3.** On that note, why is it called "electron-deficient alkenes"? Why not "unsaturated carbonyls"? Reaction with DTT is likely on the oxygenated group, rather than on the C=C double bond. Also, from a fundamental perspective, does the C=C double bond need to adjacent to the C=O group? Have the authors investigated, say, 3-pentenal?

**Response:** Not all unsaturated carbonyls are reactive to DTT. The reactivity of an alkene is determined by the contribution of all substituents. If conjugated carbonyls have electron-donating groups such as alkyl, alkenyl, and ester group, they are not anymore electron-deficient alkenes and their reactivity with DTT is low. Thus, we prefer to use electron-deficiency of alkenes (Nair et al.) for their reactivity to DTT.

Reference: Nair, D. P. et al., The thiol-michael addition click reaction: a powerful and widely used tool in materials chemistry, Chem. Mater., 26, 724-744, 2014.

**4.** Also, if acrolein indeed contributes high DTT activity, I would imagine that SOA from 1,3-butadiene would have very high DTT activity. Have the authors done any experiments to suggest that is the case?

**Response:** Acrolein was tested to describe the reactivity of functional group in organic compound. Actually, acrolein may not be present in aerosol due to their high volatility. The SOA yield of 1,3-butadiene is as low as 0.4-0.5% for 1.0-2.2 ppm 1,3-butadiene (Angove et al.). In addition, the major aerosol products (75-80%) of 1,3-butadiene are not electron-deficient alkenes, based on the study of Angove et al.

Reference: Angove, D.E. et al., The characterization of secondary organic aerosol formed during the photodecomposition of 1,3-butadiene in air containing nitric oxide, Atmospheric Environment, 24, 2006

**5.** Another suggestion for additional experiments (optional but would really strengthen the argument): if the authors can pass ozone over the SOA (e.g. collected on the filter), the ozone should selectively remove the double bonds. This way the authors can isolate the contribution to DTT from these electron deficient alkenes. While this may create additional organic hydroperoxides, the authors can easily correct for that increase by measuring total OHP with the NPBA assay.

**Response:** The electron-deficiency of alkenes increases the reactivity to DTT while electron-deficiency can significantly reduce reactivity to ozone. In addition, the products can be off-gassing from the filter by passing ozone through the filter and increase artifacts in chemical assays. Some products that have weak reactivity to DTT may become reactive to DTT due to the ozone reaction.

**6.** Does the UNIPAR model describe the abundance of electron deficient alkenes under low-NOx? I would imagine under high NOx there would be more electron deficient alkenes where C-C bond scission and ring-opening reactions are dominant, but the "unaccounted" DTT activity seems to be higher under low NOx for toluene (from Fig. 6).

**Response:** In order to response to reviewer, the quantity of conjugated compounds was estimated using the UNIPAR model. The quantity of conjugated compounds in low-NOx toluene SOA is similar to or slightly more than that in high-NOx toluene SOA. Within the error bar, the "unaccounted" DTT activity of low-NOx toluene SOA was close to that of high-NOx toluene SOA (Fig. 6 in the old version, Fig. 5 in revised manuscript).

7. I understand that the proposal about measuring DTTm as a proxy for maximum capacity (described in Section 3.1) is useful for a chemist. But I wonder what the biological relevance is. What is the lifetime of particles deposited in the lungs? I am not a lung expert, but I wonder if mucociliary clearance would make the lifetime shorter than 2 hours, and therefore DTTm at 2 h may not be that relevant.

**Response:** The DTT assay enables the evaluation of aerosol reactivity to a sulfhydryl group using a small amount of aerosol mass and simple chemical procedures, and illustrates of the potential toxic mechanisms of aerosols. However, chemical assay is limited to a screening method and the results of chemical assay need to be compared with *in-vitro* and *in-vivo* studies. According to the study of Oberdorster et al. (1988), the lung clearance of particles ranges from several minutes to one day, depending on the solubility, particle size, etc. Binding with and to tissue and cellular components can increase the retention halftimes of particles to days and months. Also, the lifetime of particles in biological systems can also be affected by the lung diseases. The clearance of particles can be retarded if the lung is not healthy.

Reference: Oberdörster G., Lung Clearance of Inhaled Insoluble and Soluble Particles, Journal of Aerosol Medicine, 1, 1988.

8. For DTTm to be meaningful, DTT must be in large excess compared to amount of SOA. What is the estimated ratio of DTT to SOA (100 uL of DTT, 700 uL of SOA solution)? Is DTT in large excess? This is not an issue when looking at the catalytic activity, because both DTT and quinones can be regenerated through catalytic cycles. But this issue arised when it is a one-step reaction with these particulate oxidizers.

**Response:** The excess amount of DTT was used to determine oxidative potential of SOA. A sentence was added and reads now,
"To ensure the pseudo-1st-order reaction between DTT and redox-active species in SOA, the SOA mass applied to DTT assay was constrained to ensure that the DTT consumption remained less than 50 % of the initial DTT concentration."

9. If hydroperoxides (and PANs) are indeed unstable at room temperature (as suggested in Section 2.3.2), did the decrease in ROOH amount correspond to a decrease in DTTt or DTTm?

**Response:** The QA/QC of chemical assays used in this study was reorganized and included in the supporting information (Section S4). The stability of $H_2O_2$ and peracetic acid with storage time was measured by NPBA assay (Fig. S8) and by DTT activity (Fig. S4). When the solution was kept at 4 $^o$C over a 6-day period, the decrease of DTT response of peracetic acid was 22% and the decrease of NPBA assay was 11%. However, SOA samples of this study were analyzed within 24 h after collection. The decrease of DTT response and NPBA response of peracetic acid stored at 4 $^o$C were insignificant compared to those of the fresh solution. Thus, we think that instability of organic peroxides will not influence of our chemical analysis. The QA/QC information of the stability of compound was shown below:

**The stability of peroxides.** To investigate the impact of the stability of organic compounds on DTT actitivities, the DTT consumptions ($\Delta$DTT at reaction time=100 min) by 400 $\mu$L $H_2O_2$ (100 $\mu$M) and 300 $\mu$L peracetic acid ($CH_3C(O)OOH$; 100 $\mu$M) were measured as a function of storage time. As shown in Fig. S4, $\Delta$DTT of $H_2O_2$ and $CH_3C(O)OOH$ stored at room temperature (RM) decreased upto 80 % and 36 % after an 8-day storage, respectivley. $\Delta$DTT of $CH_3C(O)OOH$ stored at 4 $^o$C also decreased 22% after a 6-day storage.

Relative DTT response to fresh sample

[Figure]

Figure S4. The DTT responses of $H_2O_2$ and peracetic acid stored at room temperature (RM) or 4 $^o$C. The errors was estimated using the standard deviation of three replicates.

**The stability of peroxides.** To investigate the impact of the stability of peroxides on NPBA assay, the NPBA repsonses for 400 $\mu$L $H_2O_2$ (100 $\mu$M) and 300 $\mu$L peracetic acid (100 $\mu$M) were

measured as a function of storage time. Fig. S8 illustrates that the NPBA response of $H_2O_2$ and $CH_3C(O)OOH$ stored at room temperature decreased upto 70 % and 27 % after an 8-day storage, respectivley. The decrease of the NPBA response of $CH_3C(O)OOH$ stored at 4 ºC was similar to that stored at room temperature.

[Figure]

Relative NPBA response to fresh sample

Figure S8. The NPBA responses of $H_2O_2$ and peracetic acid stored at room temperature or 4 ºC. The error bar was estimated using the standard deviation of three replicates.

10. There are large fluctuations in experiment temperatures (more than 20K). Are there systematic differences in chemical composition?

**Response:** Thanks much for your comment.  The UF-APHOR chamber uses the ambient sunlight and meteorological conditions.  Thus, the temperature profile for each experiment has a diurnal pattern.  Although temperature can influence the chemical composition of SOA due to the impact of temperature on the gas-particle partitioning of organic products, temperature is less influential on DTTt than aging time or NOx levels.  For example, DTTt values of toluene were modulated mainly by aging time (Fig. 3 in old version, Fig. 2 in revised manuscript).  For isoprene SOA, DTTt was affected mainly by NOx conditions.  We included the discussion of temperature on DTT activity in the revised manuscript (Section 3.1).

"The $DTT_t$ values of isoprene SOA was, however, higher than those observed by Tuet et al. (Tuet et al., 2017b) and Kramer et al. (Kramer et al., 2016). This difference might be caused by the degree of aerosol aging under different $NO_x$ conditions, initial OH radical sources, humidity and temperature."

Reference: Tuet, W. Y., Chen, Y., Xu, L., Fok, S., Gao, D., Weber, R. J., and Ng, N. L.: Chemical oxidative potential of secondary organic aerosol (SOA) generated from the photooxidation of biogenic and anthropogenic volatile organic compounds, Atmos. Chem. Phys., 17, 839-853, 2017b.
Kramer, A. J., Rattanavaraha, W., Zhang, Z., Gold, A., Surratt, J. D., and Lin, Y.-H.: Assessing the oxidative potential of isoprene-derived epoxides and secondary organic aerosol, Atmos. Environ., 130, 211-218, 2016

**Minor comments:**
**1.** Section 2.2: Please provide define PILS and provide details of the PILS method. Was steam used to grow particles for impaction and collection? If so, is there concern about the high temperatures leading to decomposition of thermally labile compounds important for oxidative potential?

**Response:**
We added the principal of PILS in Section 2.2 and read now,
"The aerosol particle that enters the PILS grows quickly into a droplet under the supersaturated environment and this droplet is collected on the plate by impaction."

It is possible that high temperature leads to the decomposition of thermally unstable compounds. However, the residence time of a particle inside PILS is approximately 1 second. In addition, the rapid cooling of water steam by the ambient air sample gives the high supersaturation. The resulting temperature downstream PILS is nearly ambient temperature.

**2.** Section 2.3.1 Line 1: it is unclear what "two steps" mean in parentheses. Perhaps re-word to "The two-step DTT assay: : :"

**Response:** This paragraph was modified and reads now
"DTT assay was employed to quantify the oxidative potential of SOA (Cho et al., 2005; Jiang et al., 2016). In the first step (DTT oxidation), a mixture of 700 µL SOA-PILS sample, 200 µL potassium phosphate buffer (2 mM, pH=7.4) and 100 µL DTT (1 mM) was incubated at 37 °C in a sonicator (FS30H Ultrasonic Cleaner, Fisher Scientific). For the second step (determination of the remaining DTT), the reaction between DTT and SOA was quenched by adding 1 mL 1% w/v trichloroacetic acid (a commonly used quencher of thiol oxidation)."

**3.** Section 2.3.1 Line 2: What does SOA sample mean? Is this the aqueous solution?

**Response:** "SOA sample" was changed to "SOA-PILS sample" that was the aqueous solution collection by PILS.

**4.** Section 2.3.1 Page 6 Line 1: What is the pH of the PBS?

**Response:** The information of pH=7.4 was added to the sentence.

**5.** Section 2.3.1 Page 6 Line 1: "shaken in a sonicator" is confusing. Particularly since in the later section "shaking" was compared to "sonicating".

**Response:** This sentence was modified and reads now,
"In the first step (DTT oxidation), a mixture of 700 µL SOA-PILS sample, 200 µL potassium phosphate buffer (2 mM, pH=7.4) and 100 µL DTT (1 mM) was incubated at 37 °C in a sonicator (FS30H Ultrasonic Cleaner, Fisher Scientific)."

**6.** Section 2.3.2: Can 4-nitrophenol be formed in high NOx SOA from toluene? If so, are there any negative controls?

**Response:** 4-Nitrophenol may be found in toluene SOA. The influence of 4-nitrophenol in SOA on NPBA assay was discussed in Section S4.2 (revised supporting information) and reads now,
"4-Nitrophenol, NPBA product, can also be found in toluene SOA and potentially influences on NPBA data. However, the light absorption of the SOA sample collected using PILS was negligible

at 406 nm, suggesting that NPBA data is not influenced by the light absorbing materials in SOA products."

**7.** Section 3.1 heading: remove period after title

**Response:** The period mark after title was removed.

**8.** Section 3.3 Line 5: Grek et al. does not provide any evidence that PANs can oxidize sulfhydryl groups. They only include peroxynitrites as an RNS, but PANs are peroxy nitrates.

**Response:** For the reaction of PAN with a sulfhydryl group, we cited other references. The new citations are shown below:
Products of the Reaction of Peroxyacetyl Nitrate with Sulfhydryl Compounds, Mudd et al., Archives of Biochemistry and Biophysics, 132, 1969.
Reaction of Peroxyacetyl Nitrate with Glutathione, J. B. Mudd, The journal of Biological Chemistry, 241, 1966.f

**9.** Section 4 Line 25: remove (DTT modulator: quinones)

**Response:** "(DTT modulator: quinones)" was removed from this sentence.

**10.** Section 4 Page 14 Line 8: An assay that is not water-based may not be relevant, since cells are water-based

**Response:** This sentence was deleted.

**11.** Table 1 and throughout manuscript: Does HC/NOx ratio refer to the initial or average HC/NOx? If it is the average, over what time range is the HC/NOx calculated?

**Response:** The $HC/NO_x$ ratio mentioned throughout the manuscript refers to the initial $HC/NO_x$. The title "$HC/NO_x$ (ppbC/ppb)" in Table 1 was changed to "Initial $HC/NO_x$ (ppbC/ppb)". The detection of the $HC/NO_x$ ratio over the course of experiment is complex due to the complexity in chemical species and the wall process of organics, nitric acid, and organonitrates.

**12.** Table 1 Mid-collection time: is this local time?

**Response:** "Mid-collection time" refers to the local time in Gainesville, Florida. This was clarified in the revised manuscript.

**13.** Figure 4: why is one data series labeled with experiment date, but others are not?

**Response:** The date information was removed. The initial HC/NOx ratio was added.

**Response to comments from referee #3 (Manuscript Ref. NO: acp-2017-214)**

Huanhuan Jiang et al.

mjang@ufl.edu

We really appreciate the referee for providing these constructive comments. The detailed responses are presented as below.

This study investigated the DTT activity of different types of SOA (toluene, TMB, isoprene, a-pinene) formed in the presence of NOx. Experiments are conducted in an outdoor chamber facility with different levels of NOx. Results showed that in the presence of relatively higher levels of NOx, DTT activity of toluene SOA is 2-5 times higher than other SOA. Isoprene SOA has lower DTT with increasing NOx. Other SOA appears to be insensitive to NOx. The results are discussed in the context of different DTT modulator compounds.

This is an interesting study and will be of interest to the greater research community. I have a number of questions regarding the experimental design/protocols and how the data are interpreted. Further, the conclusions need to be better justified.

The experiments are conducted with very high levels of hydrocarbon and NOx. It is not clear what the NOx levels in this study mean in terms of the reaction pathways. It appears that the authors assume that under the "low-NOx" conditions in this study, the dominate fate of the peroxy is reaction with HO2. It is not immediately clear how this is the case, especially with the high levels of hydrocarbon and NOx, and without an external HO2 source. To what extent does RO2+HO2 proceed in these experiments and whether organic hydroperoxide should be produced in the first place?

**Response:** Please find the response to question 11 (a) below.

The results in this study should be compared with prior literature and discussed. For instance, Kramer et al. (2016) found that for isoprene SOA, high-NOx conditions produce SOA that is more oxidizing compared low-NOx conditions. But, the results in this work suggest the opposite. Results from SOA formed from different hydrocarbons should also be compared to prior literature when available (e.g., Rattanavaraha et al., 2011, Tuet et al., 2017a; Tuet et al., 2017b). Tuet et al. found the DTT activity of different SOA to be fairly insensitive to RO2 fate.

**Response:** Please find the response to question 7(b).

Further, the same authors of this manuscript published a recent paper in Atmos Environ (Jiang et al. 2016) on similar experiments, but the results under similar reaction conditions are not the same as those reported in this study. Please clarify and discuss accordingly.

**Response:** Please find the response to question 11(c).

It was noted that the SOA yields in this study are consistent with prior studies. SOA yields should be compared in the context of organic mass loadings. It does not appear that the yields in this work are in line with literature. Based on the numbers provided in Table 1, the SOA yields calculated are very different from prior studies. Please see detailed comments below. The implications of these differences should be considered and discussed. If the yields are so different, what does it imply regarding their corresponding SOA composition and their health effects? On a related note, the organic mass loadings (one can calculate them based on Table 1) are high in these experiments, which will result in partitioning of more volatile species into the particle phase compared to ambient. The implications of this on the measured DTT activity in this study and how they can be applied to ambient should also be discussed.

**Response:** Please find the response to question 6(a), 6(b) and 11(d).

The authors attributed the difference in [OHP]m between low and high NOx experiments to sample collection times and associated aging. I do not think this is well-justified. Many other parameters are also changing at the same time.

**Response:** Please find the response to question 11(d).

Finally, the basis for comparison of DTT activity is time of the sample collection in many cases. Is that chosen to represent OH exposure? But if so, it is not clear that [OH] are constant and comparable across different experiments.

**Response:** Please find the response to question 11(d).

Overall, I recommend publication with major revisions. More detailed comments below.

**Major comments:**
1. Page 1, line 14. Define clearly what is considered as high vs. low NOx conditions in this study.

**Response:** The high NOx condition represents the initial HC/NOx ratio<10 ppbC/ppb. The low NOx condition represents the initial HC/NOx ratio >10 ppbC/ppb. We clarified the definition of high and low NOx conditions in the revised manuscript (Section 2.1, Figure 2 and Figure 5).

2. Page 4, methods.
a. Was seed aerosols used in the experiments? Please state clearly.

**Response:** We added one sentence to the revised manuscript (Section 2.1): "No seed aerosols were added in this study."

b. The NOx levels in this study were very high in all experiments. The authors need to define LNOx and HNOx clearly in the methods.

**Response:** Please find the response to question 1.

c. HONO was used only in toluene experiments, why? The other experiments used NOx (was that NO or NO2? Or both? In what ratio?)

**Response:** The HONO was injected during toluene experiments to accelerate the photooxidation of toluene. NO (2% in $N_2$, Airgas) was injected as NOx source in other experiments. We clarified the injection of HONO and NOx in the revised manuscript (Section 2.1), and read now,
"HC and NO (2% in $N_2$, Airgas) were injected to the chamber before sunrise. For photooxidation experiments of toluene, HONO generated from the reaction of 0.1 M $NaNO_2$ solution and 10 % w/w $H_2SO_4$ solution was injected into the chamber as a source of OH radicals."

d. The authors should provide figures (in the SI) to show the typical time series of hydrocarbon, NO, NO2, ozone, SOA mass for an experiment using HONO vs an experiment using NOx. This is critical to give some context regarding the conditions under which the SOA samples were collected for DTT analysis. (for instance, what was the NOx levels when the SOA samples were collected? This has important implications on the SOA composition and whether the results from different SOA samples can be directly compared.)

**Response:** The reaction of toluene with OH radicals is much slower than other VOCs. Thus, we added HONO to the chamber at the beginning of the experiment. All toluene experiments were conducted in the presence of HONO except one experiment on Feb/12/2016. For the consistency in toluene SOA data, the toluene experiment without HONO has been removed in the revised manuscript. In order to response to the reviewer, the time profiles of NO, NOx, HC, ozone, and SOA mass concentrations have been included in supporting information (Figure S1). SOA samples were collected when the SOA mass concentration reached nearly to the maximum point.

[Figure]

Figure S1. The time profiles of SOA mass concentration, HCs, $NO_x$, NO and $O_3$ mixing ratios through the experiments: (a) toluene (HC/$NO_x$=9 ppbC/ppb); (b) toluene (HC/$NO_x$=22 ppbC/ppb); (c) TMB (HC/$NO_x$=6 ppbC/ppb); (d) TMB (HC/$NO_x$=19 ppbC/ppb). EST, Eastern Standard Time. The shaded areas indicate the time periods of aerosol sampling.

e. What is the collection efficiency of the PILS? How is it calculated? How is it validated? As the DTT values was normalized by mass, the authors need to provide more details to justify the accuracy of the SOA mass values used in the DTT calculations.

**Response:** The collection efficiency of PILS for was published by Orsini et al (2003). We clarified the collection efficiency in the revised manuscript (Section 2.2) as below:
"The sampling efficiency of PILS is greater than 95 % for particle size ranging from 0.03 to 6 µm (Orsini et al., 2003)." A value of 95% was used as the collection efficiency of PILS.

3. Page 6, line 17. It was noted that "The SOA mass applied to DTT assay was constrained to ensure that the DTT consumption remained less than 50 % of DTT0." Why?

**Response:** We modified this sentence as follows:
"To ensure the pseudo-1st-order reaction between DTT and redox-active species in SOA, the SOA mass applied to DTT assay was constrained so that the DTT consumption remained less than 50 % of the initial DTT concentration."

4. Page 6, line 20 onwards, the NPBA assay. It is not clear how the authors treat (or correct for) potential interference from other SOA components in the NPBA assay measurements.
a. It was noted that absorbance at 406 nm (authors referring to 4-nitrophenol) of SOA before reaction with NPBA was negligible. I do not understand why this is the case. The oxidation of aromatic compounds (such as toluene) by OH in the presence can result in the formation of 4-nitrophenol? If so, why is the absorbance at 406nm negligible?

**Response:** 4-Nitrophenol can be found in toluene SOA. The influence of 4-nitrophenol in SOA on NPBA assay was discussed in Section S4.2 (revised supporting information) and reads now,
"4-Nitrophenol, a NPBA assay product, can also be found in toluene SOA and potentially influences on NPBA data. However, the light absorption of the SOA sample collected using PILS was negligible at 406 nm, suggesting that NPBA data is not influenced by the light absorbing materials in SOA products."

b. In terms of potential interference from alcohols, it was noted that glycerol aqueous solutions were tested using the NPBA method and no measurable absorption appeared in UV spectrum. The authors then concluded that the UV absorption spectrum was originated from 4-nitrophenol. Multi-functional alcohols can be formed from the oxidation of hydrocarbons. How do the authors justify that the results from glycerol is representative and that the measured absorption is purely from 4-nitrophenol formed by NPBA reactions with organic peroxides, without any interference from other SOA components?

**Response:** The mechanism of NPBA assay is the production of 4-nitrophenol from the reaction between OOH functional group and NPBA. The SOA contains a lot of OH functional groups, which might also react with NPBA. To make sure that the OH functional groups don't interfere with the detection of hydroperoxides, we did the test of glycerol and found that there was no 4-nitrophenol produced from the mixture of glycerol and NPBA. The influence of multi-alcohols on

NPBA assay was moved to Section S4.2 (revised supporting information) and reads now (next page),

"**The influence of multi-alcohols and 4-nitrophenol on NPBA assay.** It has been reported that boronic acid can react with multi-alcohols to form colorful products (Kim et al., 2007). SOA products are usually multifunctional and contain multi-alcohols.  To examine the possible interference by NPBA-alcohol adducts, the glycerol aqueous solution was tested for the NPBA method.  However, no measurable absorption appeared in the UV spectrum."

c. Why "no more than 10 ug SOA was applied to the NPBA assay"?

**Response:** We removed this sentence in the revised manuscript.

5. Page 7, line 16 onwards, PAN analysis. Please provide further details to show that PAN are completely hydrolyzed in 15 mins.

**Response:** The information of hydrolysis time for Griess assay was included in supporting information (S4.3) and reads now,

**"Hydrolysis time of PAN products in SOA.** In this study, the hydrolysis time for PANs was set to 15 min.  To ensure that 15 min is enough to hydrolyse the PANs in SOA products, the Griess response with 15-min hydrolysis was compared to that with 30-min hydrolysis.  There was no significant difference in Griess response between a 15-min hydrolysis and a 30-min hydrolysis.  "

6. Page 8, line 14-18. SOA yields from different hydrocarbons cannot be compared without the context of organic aerosol mass, as yield is typically a function of organic aerosol mass (Odum equation).
a. In Table 1, while one can calculate the organic mass loading from ΔHC and yield, the authors should also provide the organic mass loadings in the table to facilitate easier comparison of yields from this study to prior studies.

**Response:** The SOA mass loadings were included in Table 1. The new table is shown in the next page:

Table 1. Outdoor chamber experiment conditions

| HC and Date | Initial HC ppb | Initial NO$_x$ (HONO)[a] ppb | Initial HC/NO$_x$ ppbC/ppb | [SOA]$_{max}$[b] µg/m³ | ΔHC[c] ppb | Y % | Mid-collection time[d] | RH[e] % | Temp[e] K | Chemical assay[f] |
|---|---|---|---|---|---|---|---|---|---|---|
| ***Toluene*** | | | | | | | | | | |
| 13 Feb 2016 | 641 | 525 (193) | 9 | 229 | 403 | 15.1 | 13:40 | 22–63 | 281–303 | DTT |
| 01 May 2016 | 935 | 766 (133) | 9 | 348 | 631 | 14.6 | 14:20 | 18–46 | 294–316 | DTT, PAN |
| 01 May 2016 | 938 | 301 (73) | 22 | 292 | 542 | 14.3 | 12:10 | 21–48 | 294–315 | DTT, PAN |
| 23 May 2016 | 691 | 906 (250) | 5 | 148 | 546 | 7.1 | 13:20 | 18–60 | 288–315 | DTT, Enhance |
| 23 May 2016 | 735 | 313 (86) | 16 | 147 | 421 | 9.3 | 15:40 | 15–22 | 307–316 | DTT, Enhance |
| 18 Aug 2016 | 640 | 783 (179) | 6 | 178 | 517 | 9.1 | 12:30 | 24–61 | 297–319 | DTT, OHP |
| 06 Aug 2016 | 610 | 240 (55) | 18 | 75 | 216 | 9.2 | 12:30 | 43–59 | 297–305 | DTT |
| 18 Aug 2016 | 342 | 107 (24) | 22 | 44 | 227 | 5.2 | 14:20 | 20–38 | 303–321 | OHP |
| 17 Nov 2016 | 622 | 179 (43) | 24 | 139 | 452 | 8.1 | 13:20 | 12–56 | 282–309 | DTT[g] |
| ***TMB*** | | | | | | | | | | |
| 04 Oct 2015 | 613 | 920 | 6 | 201 | 613 | 6.7 | 14:40 | 20–43 | 290–310 | DTT |
| 04 Oct 2015 | 657 | 310 | 19 | 207 | 542 | 7.8 | 13:20 | 24–46 | 290–306 | DTT |
| 20 Feb 2016 | 589 | 1024 | 5 | 150 | 548 | 5.6 | 13:00 | 14–60 | 282–311 | DTT |
| 20 Feb 2016 | 583 | 156 | 34 | 128 | 455 | 5.7 | 14:40 | 16–61 | 282–311 | DTT |
| 11 Jan 2016 | 595 | 256 | 21 | 114 | 414 | 5.6 | 15:50 | 23–81 | 274–298 | Enhance |
| ***Isoprene*** | | | | | | | | | | |
| 23 Apr 2016 | 2693 | 2680 | 5 | 352 | 2693 | 4.7 | 12:00 | 18–48 | 290–314 | DTT |
| 23 Apr 2016 | 2755 | 430 | 32 | 93 | 2755 | 1.2 | 13:30 | 23–51 | 290–312 | DTT |
| 14 May 2016 | 2928 | 2800 | 5 | 406 | 2928 | 5.0 | 14:20 | 17–47 | 292–315 | DTT, Enhance |
| 14 May 2016 | 2858 | 423 | 34 | 107 | 2858 | 1.3 | 12:00 | 25–55 | 293–312 | DTT |
| 22 Jul 2016 | 2525 | 2423 | 5 | 246 | 2525 | 3.5 | 13:20 | 20–55 | 297–320 | PAN (gas)[h] |
| 22 Jul 2016 | 2718 | 473 | 29 | 70 | 2718 | 0.9 | 12:50 | 23–58 | 297–320 | PAN (gas)[h] |
| 20 Aug 2016 | 3060 | 3300 | 5 | 279 | 3060 | 3.3 | 12:30 | 20–58 | 296–321 | DTT, OHP, PAN |
| 20 Aug 2016 | 3173 | 583 | 27 | 125 | 3173 | 1.4 | 11:50 | 25–61 | 297–318 | DTT, OHP, PAN |
| ***α-Pinene*** | | | | | | | | | | |
| 25 Feb 2016 | 319 | 639 | 5 | 257 | 319 | 14.5 | 15:00 | 21–63 | 278–299 | DTT |
| 25 Feb 2016 | 323 | 91 | 36 | 650 | 323 | 36.1 | 13:30 | 25–67 | 278–298 | DTT |
| 18 Jan 2016 | 257 | 144 | 18 | 223 | 257 | 15.6 | 15:50 | 25–78 | 275–297 | Enhance |

[a] For toluene experiments, NO$_x$ was contributed by NO, NO$_2$ and HONO. The concentration of HONO was estimated using the difference in the NO$_2$ signal with and without the base denuder (1 % Na$_2$CO$_3$+1 % glucose).

[b] [SOA]$_{max}$ is the maximum SOA concentration during the aerosol collection.

[c] ΔHC is the consumption of HC when the SOA concentration reached to a maximum during the aerosol collection.

[d] This column is the mid-collection time (based on the Eastern Standard Time (EST)) of SOA sampling.

[e] The RH and temperature conditions shown in the Table 1 were recorded from the beginning of photooxidation (sunrise) until the ending of PILS sampling.

[f] The SOA samples were applied to a series of chemical assays, namely DTT assay (DTT), DTT enhancement (Enhance), organic hydroperoxides analysis (OHP), and PAN analysis (PAN).

[g] For DTT measurement of toluene SOA sample collected on 17 Nov. 2016, the concentration of potassium phosphate buffer (0.8 mM) in the first step of DTT assay was two times higher than the typical buffer concentration (0.4 mM). The DTT$_m$ of the toluene SOA sample (17 Nov. 2016) is shown in Fig. 3.

[h] The concentration of gaseous PAN products (collected by an impinger) was measured by the Griess assay.

b. I quickly did the calculations, but found that the yields for the different hydrocarbons are very different from the previous studies cited in the manuscript. For instance, the yield for isoprene SOA photooxidation (with NOx/isoprene ratio of 5) is all 5% for Kroll et al., Xu et al., and this study. However, the organic mass loading that corresponds to this 5% SOA yield in Kroll et al and Xu et al is an order of magnitude smaller than this study. This means that when plotted in the Y vs. Mo (Odum equation) space, the SOA yields under this specific NOx/isoprene ratio in this study is substantially lower than all previous studies. Discrepancies also exist for other hydrocarbons. The authors should conduct a detailed comparison with previous studies by showing their data in the yield curve space and comparing with others. The discrepancies should be discussed.

**Response:** Thanks for point out this comparison.
**Low NOx**: The SOA yields of low-NOx isoprene in this study were lower than the values reported by Kroll et al. (2006) or Xu et al. (2014) by several reasons. The HC ppbC/NOx ratio of our study (low $NO_x$: 27-34) was much different from the study by Kroll and Xu ($NO_x$-free). The maximum temperature of the experiments in this study (312-320 K) was higher than the temperature condition in the study of Kroll (around 298 K) or Xu (around 298 K). The temperature effect on SOA yields were discussed in the revised manuscript (Section S3.1) and reads now,
"Our SOA yields for isoprene SOA were lower than those reported in previous studies (Carlton et al., 2009; Xu et al., 2014) because the temperatures in our studies were higher than those sourced from indoor chambers."

In order to response to the reviewer, our isoprene SOA yields at high NOx conditions (HC/NOx = 5) were compared to the theoretically predicted SOA yields using the two-product model (Carlton et al. 2009).

$$Y_{\text{high NOx}}=\frac{0.154}{(K_{\text{OM,high NOx}}(\text{T})M_0)^{-1}+1}$$

$$K_{\text{OM,high NOx}}(\text{T})=K_{\text{ref}}\times\exp[\frac{\Delta H_{\text{vap}}}{R}\left(\frac{1}{\text{T}}-\frac{1}{\text{T}_{\text{ref}}}\right)]$$

$K_{\text{ref}}$ =0.0020, $T_{\text{ref}}$=303K, $M_0$ is the total aerosol concentration ($\mu g/m^3$). At T=313K (a typical temperature during our isoprene experiments), $K_{\text{OM,high NOx}}$=0.00117. The observed SOA yields agrees well with the predicted yields as shown below,

[Figure]

Reference: Carlton et al., A review of Secondary Organic Aerosol (SOA) formation from isoprene, Atmos. Chem. Phys., 9, 2009

7. Page 9, line 8-15. Comparison of response of aerosols from different hydrocarbons. This section needs to be expanded to include more discussions (in addition to the description of results).

a. The authors noted that a previous study by Fujitani et al. (2012) with epithelial cells is consistent with this study. Can the cellular results from Fujitani et al. (2012) be directly compared to the DTT in this study? Please discuss.

**Response:** Because no cellular study was conducted in this paper, we removed the comparison of our results with those of Fujitani et al. In order to response to the reviewer, a new citation was included in the revised manuscript, and reads now,

"The DTT activity of a given SOA can be applied to the assessment of SOA's ability to oxidize cellular materials. For example, the recent study by Tuet et al. (Tuet et al., 2017a) reported a positive nonlinear correlation between DTT activities and ROS production in murine alveolar macrophages."

b. How do results in this work compare to previous studies? For instance, recent work from Tuet et al. showed that the response from naphthalene is higher than other hydrocarbons such as m-xylene and a-pinene; McWhinney has also demonstrated previously that showed naphthalene SOA is highly redox-active. It would be useful that the authors provide some context and discuss the DTT activity of different SOA with respect to previous work.

**Response:** In the previous manuscript, the DTT activity of naphthalene SOA reported by McWhinney was discussed in the Section 1. The DTT activities of SOA in this study were also further compared with other studies (Sect. 3.1, revised manuscript) and reads as:

"The $DTT_t$ values of this study were also compared with those reported in previous studies. The $DTT_t$ values of α-pinene SOA in this study were close to those reported by Tuet et al. (Tuet et al., 2017b). The $DTT_t$ values of isoprene SOA was, however, higher than those observed by Tuet et al. (Tuet et al., 2017b) and Kramer et al. (Kramer et al., 2016). This difference might be caused by the degree of aerosol aging under different $NO_x$ conditions, initial OH radical sources, humidity and temperature."

8. Page 9, line 25. It was stated that ": : :DTTm of low-NOx isoprene SOA was much higher than that of high-NOx isoprene SOA". This does not seem like it is the case from Figure 4. Some data points overlap and are within uncertainty.

**Response:** We re-draw this figure to make the difference of DTTm between low-NOx isoprene and high-NOx isoprene clearer. As shown in Figure 3 (revised manuscript), it is obvious that the DTTm of low-NOx isoprene SOA was significantly higher than high-NOx isoprene SOA after a long-time reaction (t=120 min).

[Figure]

Figure 3. Time profile of $DTT_m$ for toluene and isoprene SOA under different $NO_x$ conditions. To achieve the completion of the reaction between DTT and SOA, the $DTT_m$ of toluene sample (initial $HC/NO_x$=24 ppbC/ppb collected on 17 Nov. 2016) was measured with a 0.8 mM potassium phosphate buffer in the first step of DTT assay (2 times higher than the typical buffer concentration (0.4 mM)). Each error bar was calculated by $t_{0.95} \times \sigma/\sqrt{n}$ using three replicates, where $t_{0.95}$ is the t-score (4.303 for $n = 3$ replicates) with a two-tail 95 % confidence level.

9. Page 9, line 27. (and page 10) But for the "toluene LNOx-17 Nov 2016", the data are not linear? i.e., will the data also start from the origin, if so, considering the origin and the 5 data points in Figure 4, the overall trend is then non-linear? Please discuss.

**Response:** We re-draw figure 4 by adding lines for each case. The DTTm of toluene LNOX (collected on 17 Nov 2016, $HC/NO_x$=24 ppbC/ppb) is not linear with reaction time and eventually reached to a plateau (Figure 3, revised manuscript). Please find Figure 3 in response to question 8.

10. Page 10, quinones. The authors stated that "the redox cycling of quinones was not the major mechanism underlying DTT consumption by the SOA". A small contribution of quinones to the total aerosol mass does not necessarily mean they are not important for overall toxicity? Oxidation of the aromatic compounds used in this study can lead to formation of quinones (e.g., Bloss et al., 2005). Many previous studies have pointed to the importance of quinones in ambient PM toxicity. It is not clear how the results in this work should be placed in the context of previous work that pointed to the importance of quinones for PM toxicity. Please discuss.

**Response:** In the previous manuscript (Section 1), we discussed the difference in compositions between SOA and typical combustion particulates. Combustion particles contain a small amount of metals, PAHs and oxy-PAHs, which can be redox catalyzers. Such compounds in SOA are negligible.

11. Page 11, NOx conditions.
a. Line 5 onwards. The authors noted that under low NOx conditions, RO2 predominately reacts with HO2, producing hydroperoxides (among other products). This accuracy of this statement will depend on what precisely the "low NOx conditions" are, as "low NOx conditions" do not directly (or necessarily) translate to RO2+HO2 reactions. It is not clear that RO2+HO2 is the dominant reaction under the conditions of this study. The hydrocarbon concentration used in this study is very high (hundreds of ppb to several ppm), there is abundant NOx (even under "low NOx" conditions), but no addition HO2 source (such as H2O2). With this, it is not clear how RO2+HO2 dominates. Since a large fraction of the discussions in the manuscript hinged on this, it is critical that this is justified clearly in the manuscript. The authors can perform a simple simulation of the relative importance of different RO2 reactions under their "low" and "high" NOx conditions.

**Response:** Please find the definition of low-NOx and high-NOx in response to question 1. We clarified the NOx condition in Figures 2 and 5. As suggested by the reviewer, we simulated the relative significance of $RO_2+NO$ and $RO_2+HO_2$ at two different HC (ppbC)/$NO_x$ ratios (6 and 25) for toluene. Based on integrated reaction rate (IRR) analysis for several $RO_2$ species, the reaction of $RO_2+NO$ is very sensitive to $NO_x$ condition: i.e., the IRR at the high $NO_x$ level is 5 times higher than that at the low $NO_x$. The IRR of $RO_2+HO_2$ is higher at the low $NO_x$ level but relatively less sensitive than the reaction of $RO_2+NO$. However, our IRR analysis is for the gas phase oxidation. The composition of aerosol is complex due to the involvement of aerosol phase reactions of various multifunctional organic compounds.

b. The authors shall compare and discuss their results in the context of previous studies. For instance, a recent Kramer et al. (2016) concluded that High-NOx conditions produce SOA that is more oxidizing compared low-NOx conditions. This results from this work showed the opposite. Please discuss. Also, Tuet et al. (2017a, 2017b) specifically studied the toxicity of SOA (including isoprene) under RO2+HO2 vs RO2+NO reactions conditions, and found DTT activity of isoprene SOA to be similar under RO2+HO2 and RO2+NO conditions, also for other SOA except naphthalene.
**Response:** Please find the response to question 7(b).

c. How do the data in this study compare the results from a previous study by the same author (Jiang et al. AE, 2016). For example, the "without denuder" data in Figure 2 Jiang et al AE paper can be compared to those in the current study. But comparing this study and their previous AE publication, the results (in terms of the DTTt values) are different for each hydrocarbon under similar conditions? Please compare and discuss, and confirm self-consistency if that is the case.

**Response:** The DTTt values reported in this study should be compared to those with denuder in our previous study (Jiang et al. AE, 2016), because the carbon denuder was also applied in this study (Section 2.1). The effect of denuder on DTTt has been explained in our previous study. Overall, there was no difference between the DTTt reported in this study and the values reported by our previous study (Jiang et al. 2016). For example, the DTTt of isoprene SOA (low NOx) was about 50-60 pmol min$^{-1}$ µg$^{-1}$ in this study, which was the same to the values reported before. The DTT$_t$ of toluene SOA (with denuder, collected during 13:30-14:50) was reported to be 43 min$^{-1}$ µg$^{-1}$ in the previous study, and was lower than the DTTt of toluene SOA (collected in the morning or early afternoon) due to the aging effect. The influence of aging on DTT response was consistent with that reported in this study (Section 3.1).

d. Line 20, and figure 6. What is the x-axis (time) supposed to be a surrogate of? If it is supposed to be a surrogate for OH exposure, then the OH level should be the same is each experiment. Is this the case? The authors simply explained the difference in [OHP]m between low and high NOx toluene SOA as the low-NOx SOA being collected at a later time and resulted in a lower level of [OHP]m due to further reactions/photooxidation. I do not think the authors can discount other factors, such as the varying RH (maybe SOA composition is different due to different RH?), organic mass loading (when I used the numbers in Table 1 and calculated the mass loadings for the toluene experiments shown in Figure 6, the loadings are very different for the two experiments), etc?

**Response:** The x-axis is based on local time (EST). In Figure 6, the RH conditions of high-NO$_x$ and low-NO$_x$ experiments for toluene SOA are similar because two experiments were conducted in the same day. As pointed by the reviewer, other factors, like organic mass loading, may also influence the SOA compositions. Overall, we think that the major factors to affect SOA compositions are the NO$_x$ condition and the aging process.

12. Table 1. Was ozone present in these experiments? If so, please include some information here. This should also be specified and discussed clearly in the manuscript, in case some of the SOA is formed from ozonolysis in addition to OH oxidation.
**Response:** Ozone was not introduced to chamber but was formed during the photooxidation of biogenic HCs (i.e. isoprene and α-pinene). The additional sentences were added to the 1st paragraph in section 3.1 and read now,

"Aromatic hydrocarbons (toluene and 1,3,5-trimethylbenzene) are mainly oxidized by OH radicals while biogenic hydrocarbons (isoprene or a-pinene) by both OH radicals and ozone. Based on the integrated reaction rate (IRR) analysis, the oxidation of isoprene by OH radicals is at least 3 times higher than that by ozone at the low NOx condition. The oxidation of biogenic hydrocarbons is dominated by OH radicals, particularly, in the morning.

**Minor Comments:**

1. Page 1, line 26. Clearly state in the abstract under what NOx conditions the sentence "The amount of organic hydroperoxide was substantial, while PANs were found to be insignificant for both SOA."

**Response:** This sentence was modified as follows

"Under the $NO_x$ conditions (HC/$NO_x$ ratio: 5-36 ppbC/ppb) applied in this study, the amount of organic hydroperoxide was substantial, while PANs were found to be insignificant for both SOA."

2. Page 1, line 29. Clearly state what "model compound study" refers to.

**Response:** This sentence was modified as follows,

"The DTT assay results of the model compounds study suggest that electron-deficient alkenes, which are abundant in toluene SOA, could also modulate DTTm"

3. Page 2, line 8. The author should also cite McDonald et al. (2012, Inhal. Toxicol), McWhinney et al. (2013, ACP), Kramer et al., (AE, 2016), Tuet et al. (2017, ACP), and Tuet et al. (2017, ACPD).

**Response:** These references were included in the revised manuscript (Section 1).

4. Page 4, line 7-9. The authors should state clearly that only selected (but not all) toluene and isoprene SOA samples are analyzed with the Griess and NPBA assays.

**Response:** This sentence was modified as follows

"Selected toluene and isoprene SOA samples were immediately applied to the DTT assay and the quantification of particulate oxidizers."

5. Page 5, line 16. Was 23 May 2016 a typo? Should it be July 22? (based on Table 1)

[revised manuscript text omitted]